# Structures of honeybee-infecting Lake Sinai virus reveal domain functions and capsid assembly with dynamic motions

Nai-Chi Chen [1,9], Chun-Hsiung Wang [2,9], Masato Yoshimura [1], Yi-Qi Yeh [1], Hong-Hsiang Guan [1], Phimonphan Chuankhayan[1], Chien-Chih Lin [1], Pei-Ju Lin[1,3], Yen-Chieh Huang [1], Soichi Wakatsuki[4,5], Meng-Chiao Ho [2] ✉ & Chun-Jung Chen [1,6,7,8] ✉

Understanding the structural diversity of honeybee-infecting viruses is critical to maintain pollinator health and manage the spread of diseases in ecology and agriculture. We determine cryo-EM structures of $T = 4$ and $T = 3$ capsids of virus-like particles (VLPs) of Lake Sinai virus (LSV) 2 and delta-N48 LSV1, belonging to tetraviruses, at resolutions of 2.3–2.6 Å in various pH environments. Structural analysis shows that the LSV2 capsid protein (CP) structural features, particularly the protruding domain and C-arm, differ from those of other tetraviruses. The anchor loop on the central β-barrel domain interacts with the neighboring subunit to stabilize homo-trimeric capsomeres during assembly. Delta-N48 LSV1 CP interacts with ssRNA via the rigid helix α1', α1'–α1 loop, β-barrel domain, and C-arm. Cryo-EM reconstructions, combined with X-ray crystallographic and small-angle scattering analyses, indicate that pH affects capsid conformations by regulating reversible dynamic particle motions and sizes of LSV2 VLPs. C-arms exist in all LSV2 and delta-N48 LSV1 VLPs across varied pH conditions, indicating that autoproteolysis cleavage is not required for LSV maturation. The observed linear domino-scaffold structures of various lengths, made up of trapezoid-shape capsomeres, provide a basis for icosahedral $T = 4$ and $T = 3$ architecture assemblies. These findings advance understanding of honeybee-infecting viruses that can cause Colony Collapse Disorder.

Honeybees play a key role, as vital pollinators, in wildlife plants and agriculture crops. *Apis cerana* Fabricius (*A. cerana*) and *Apis mellifera* Linnaeus (*A. mellifera*), commonly known as "Eastern honeybee" and "Western honeybee", respectively, are economically important farmed honeybees over 90% of honeybee agriculture industry[1,2]. One fraction of the honeybee colony has globally suddenly disappeared, which is attributed to the Colony Collapse Disorder (CCD), a serious problem to cause a loss of hives (up to 90%) in apiaries[3–5]. CCD is a phenomenon that occurs when most worker bees in a colony vanish and leave behind only a few nurse bees to care for the queen and the remaining

[1]Life Science Group, Scientific Research Division, National Synchrotron Radiation Research Center, Hsinchu 30076, Taiwan, ROC. [2]Institute of Biological Chemistry, Academia Sinica, Taipei 115, Taiwan, ROC. [3]Institute of Bioinformatics and Structural Biology, National Tsing Hua University, Hsinchu 30043, Taiwan, ROC. [4]Department of Structural Biology, Stanford University, Stanford, CA 94305, USA. [5]SLAC National Accelerator Laboratory, Stanford Synchrotron Radiation Lightsource, Structural Molecular Biology, Menlo Park, CA 94025, USA. [6]Department of Physics, National Tsing Hua University, Hsinchu 30043, Taiwan, ROC. [7]Department of Biotechnology and Bioindustry Sciences, National Cheng Kung University, Tainan 701, Taiwan, ROC. [8]Department of Biological Science and Technology, National Yang Ming Chiao Tung University, Hsinchu 30010, Taiwan, ROC. [9]These authors contributed equally: Nai-Chi Chen, Chun-Hsiung Wang. ✉e-mail: Joeho@gate.sinica.edu.tw; cjchen@nsrrc.org.tw

immature bees. The major prevalence of the CCD-affected and collapsing colonies is caused by parasites, pests, and pathogens[3,6], especially RNA-containing viruses, such as Acute bee paralysis virus (ABPV), Israeli acute paralysis virus (IAPV), Kashmir bee virus (KBV), Black queen cell virus (BQCV) and Big sioux river virus (BSRV) of the *Dicistroviridae* family, Deformed wing virus (DWV), Sacbrood virus (SBV) and Slow bee paralysis virus (SBPV) of the *Iflaviridae* family and unclassified viruses, including Chronic bee paralysis virus (CBPV), and Lake Sinai virus (LSV)[6–11]. Primarily owing to the formation of parasitic mite and virus complexes, all these viruses are responsible for a contagious and infectious disease of honeybees that can lead to high mortalities and CCD[12,13].

Honeybee-infecting LSV is currently classified into a family *Sinhaliviridae* within order *Nodamuvirales*[14,15]; however, its capsid gene and monopartite genome structure presumably are closer to betatraviruses of the family *Alphatetraviridae* and Providence virus (PrV) of the family *Carmotetraviridae* with a characteristic $T = 4$ quasi-equivalence nonenveloped capsids packaging the single-stranded positive-sense RNA genome[10,16,17]. Phylogenetic analyses of LSV based on a nucleotide comparison of RNA-dependent RNA polymerases (RdRP) indicated that there are two common phylogenetic clusters, termed LSV1 and LSV2, and several additional lineages, such as LSV3–8[10,18]. The RNA genome sizes of LSV1 and LSV2 are ~5.6 kb, encoding three major genes: the Orf1 with the unclear function, RdRP responsible for viral RNA replication, and the capsid protein (CP) for the host recognition and viral capsid assembly[10,19,20].

Viral CP plays important roles in the viral life cycle. CP forms the capsid shell that protects genomes and is responsible for both host specificity and receptor binding to transport the viral genome into infected host cells, or for intracellular trafficking. Recombinant CPs could assemble into virus-like particles (VLPs) that preserve the structural insight and antigenicity of mature infectious virions[21,22]. LSV1 and LSV2 CPs with molecular masses ~63 and 57 kDa, respectively, share low sequence similarities with other known viruses of families *Dicistroviridae*, *Iflaviridae*, and *Nodaviridae* as well as tetraviruses. No high-resolution structural information on the capsid-related organization of the LSV was available before this work.

To improve our understanding of LSV capsid architectures and their relations to the CCD disease, we purified recombinant LSV1 and LSV2 CPs, assembled the corresponding VLPs in vitro, and determined the atomic structures of capsids of four various types at three pH environments at the highest resolution of 2.3 Å: $T = 4$ and $T = 3$ LSV2 VLPs at neutral pH 7.5; $T = 4$ and $T = 3$ LSV2 VLPs at basic pH 8.5; $T = 4$ and $T = 3$ LSV2 VLPs at acidic pH 6.5 and $T = 4$ and $T = 3$ delta-N48 LSV1 VLPs at acidic pH 6.5, using cryogenic electron microscopy (cryo-EM) combined with X-ray crystallography and small angle X-ray scattering (SAXS). The structures of LSV2 and delta-N48 LSV1 VLPs differ significantly from those of other capsids of tetraviruses. Analyses of these capsid structures reveal the intermolecular interactions that drive subunits assembly into homo-trimeric capsomeres and eventually into a complete viral capsid in which specific domains, such as the N-terminal domain, the helical domain, and the β-barrel domain/C-arm, could encapsidate nucleic acids. Multiple linear structures composed of several trapezoid-shaped capsomeres that might be related to the capsid assembly were observed in cryo-EM images. The particle polymorphism of LSV VLPs, together with reversible dynamic capsid body motions, reveals the dynamical pentameric and hexameric capsomere interfaces at varied pH environments and provides structural insight into the mechanism of capsid assembly, functions, and maturation.

## Results

### Capsid architecture of LSV2 VLP

LSV2 CPs were over-expressed and purified from lysates of *Escherichia coli* (*E. coli*), and subsequently VLPs were self-assembled in vitro (Fig. 1a). The LSV2 VLPs reveal structural polymorphism of $T = 4$ and $T = 3$ particles according to EM images (Fig. 1a). From ~11,000 particle images recorded on a direct electron detector with a super-resolution mode on the FEI Titan Krios microscope, the capsid structures of LSV2 VLP within $T = 4$ and $T = 3$ architectures were determined by localized reconstructions to 2.4 and 2.6 Å resolutions on averaging around the icosahedral symmetric axes, from a respective single dataset (Supplementary Figs. 1–3 and 5–7) (Supplementary Tables 1–4). The $T = 4$ and $T = 3$ LSV2 VLPs show external diameters of ~492 and ~448 Å, and internal diameters of ~294 and ~250 Å, respectively, with a protein shell thickness of ~100 Å (Supplementary Fig. 9a).

The models of $T = 4$ and $T = 3$ LSV2 VLPs were built and refined with the homo-trimeric subunits A/B/C or D/D/D. Each icosahedral asymmetric unit (iASU) of the $T = 4$ particle consists of four subunits A, B, C, and D, whereas one iASU of the $T = 3$ particle comprises A, B, and C subunits with the C/C dimer taking up a position similar to the C/D dimer of $T = 4$ particles (Fig. 1c). Accordingly, the $T = 4$ LSV2 VLP is an icosahedral symmetrical sphere consisting of 240 copies of the 57-kDa full-length CP with 80 apparent trimeric spikes on the VLP surface, whereas the $T = 3$ LSV2 VLP is composed of 180 copies with 60 trimeric spikes. Long spikes (length 3.6 nm) protrude from the center of each homo-trimeric capsomere at icosahedral 3-fold (I3) and quasi 3-fold (Q3) axes in one $T = 4$ capsid, and at Q3 axes in one $T = 3$ capsid. Most side chains of residues of $T = 4$ and $T = 3$ LSV2 VLPs could be resolved with clear density maps (Fig. 1b), enabling constructions de novo of the accurate atomic models. Both CP models of $T = 4$ and $T = 3$ LSV2 VLPs contain residues 66–513 except for the first 65 amino acids of the N-terminal domain (NTD) and 7 amino acids of the C-terminal domain (CTD) at the end without defined density, indicating that they are flexible or disorder. The spike regions, corresponding to the major proportion of the protruding domain (P-domain), however, lack some interpretable side-chain densities, most likely because the P-domains exhibit flexibility at the outer capsid surfaces.

### Structural differences between $T = 4$ and $T = 3$ LSV2 VLPs

Besides the overall differences in VLP diameters and CP numbers, the local difference in $T = 4$ and $T = 3$ architectures is that the icosahedral 2-fold (I2) axes in $T = 4$ particles exist between subunits B, C, and D in two sets in the quasi six-fold (Q6) arrangement but I2 axes in $T = 3$ particles are present along the C/C homo-dimeric capsomeres (Fig. 1c). Within the viral capsids of both $T = 4$ and $T = 3$ LSV2 VLPs, the interfaces of two central β-barrel domains of A/B homo-dimeric capsomeres are bent ~10°, different from the flat conformations of C/D and C/C homo-dimeric capsomeres (Fig. 1e).

On the $T = 4$ LSV2 VLP, there are 30 large shaped holes of a diameter of ~22 Å at I2 axes and 12 smaller shaped holes of a diameter of ~5 Å at I5 axes (Fig. 1f). The local environment of the 20 large shaped holes at I3 axes and 12 smaller shaped holes at I5 axes of $T = 3$ LSV2 VLP is similar to that of $T = 4$ LSV2 VLP. The existence of spacious holes might provide a clue and opportunity for the viral genome release of LSV2 through an expanded viral capsid (ssRNA of a diameter of ~7 Å), whereas the smaller holes with pentameric capsomeres potentially lead to a local structural stabilization for capsid assembly.

### Structural characteristics of LSV2 CP

Each subunit of the $T = 4$ and $T = 3$ LSV2 VLPs is folded into several domains: the NTD (residues 1–65), the interior helical domain (70–80, 438–449, and 468–476), the central β-barrel domain (102–112, 150–157, 163–170, 228–243, 251–257, 322–325, 331–334, 350–356, 394–402, and 406–420), the exterior P-domain (261–301), the C-arm (477–513), and the CTD (514–520) (Fig. 1d and Supplementary Fig. 10). A segment $_{28}$RRRRNRRRRR$_{37}$ of NTD comprises the positively charged arginine-rich motif (N-ARM). The NTD of each subunit in both $T = 4$ and $T = 3$ LSV2 VLPs is disordered and suspended in the inner viral capsid.

The interior helical domain of LSV2 CP encompasses 32 amino acids consisting of three 3.5-turn helices (helices α1, α11, and α12), of

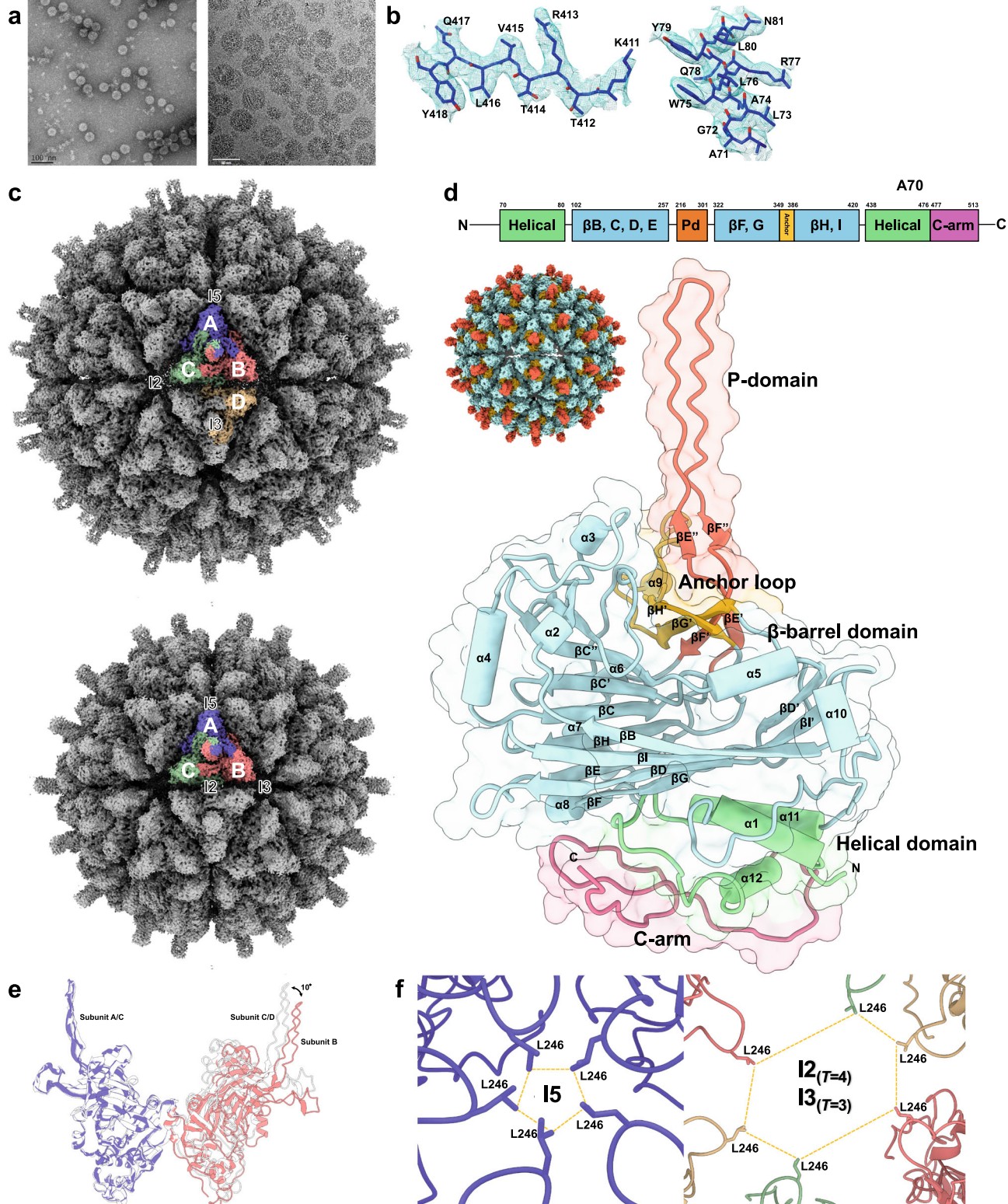

**Fig. 1 | Characterization of LSV2 VLP assembly and the CP structure. a** The represented negative-stained (left) and cryo-micrograph (right) shows LSV2 CP self-assembly in vitro. 20 independent micrographs (left) and 11,245 micrographs (right) gave similar results. Scale bars: 100 (left) and 50 nm (right). **b** Representative cryo-EM electron densities of several amino acids and the fitted atomic models. **c** The overall cryo-EM densities and structures of $T = 4$ (upper) and $T = 3$ (lower) LSV2 VLPs containing one icosahedral asymmetry unit colored in purple (subunit A), red (subunit B), green (subunit C), and yellow (subunit D, upper), respectively.

**d** A schematic diagram and structure showing the modular organization of LSV2 capsid subunit A, consisting of the interior helical domain (green), the central β-barrel domain (cyan) and the exterior P-domain (salmon red), the anchor loop (orange) and the C-arm (pink). **e** Superposition of the homo-dimeric capsomeres A/B (purple and red) with C/D and C/C (light gray) from $T = 4$ and $T = 3$ LSV2 VLPs viewed tangentially to the capsid shell. **f** The two sizes of the shaped holes along I5 (left) and I2/I3 (right) axes, respectively. All L246 residues in these subunits are shown in stick. Subunits A–D are colored as in Fig. 1c.

which α1 is from the N-terminus and α11 and α12 are from the C-terminus of the CP polypeptide, and two connecting loops (Fig. 1d). The helical domain is folded into a triangular spiral and located near the central jelly-roll β-barrel domain in $T = 4$ and $T = 3$ LSV2 VLPs; the inner capsid surfaces are filled with three sets of helices α1, α11 and α12 from one homo-trimeric capsomere located around the Q3 and I3 axes (Supplementary Fig. 11a).

The central β-barrel domain adopts a jelly-roll β-sandwich conformation with antiparallel β-strands, connected with several loops and helices of various lengths, respectively (Fig. 1d), and contributes to inter-subunit contacts for the arrangements of $T = 4$ and $T = 3$ capsid shells. Pairwise structural comparisons of the central β-barrel domains of LSV2 with several CPs of other viruses, including $T = 4$ *Nudaurelia capensis ω* virus (NωV) (PDB entry 1OHF) (Z-score 26.4)[23], $T = 3$ Pariacoto virus (PDB entry 1F8V) (z-score 20.1)[24] and $T = 1$ Infectious bursal disease virus (PDB entry 2GSY) (z-score 15.9)[25], using DALI[26], indicate that these CPs utilize a similar central β-barrel domain to form the canonical contiguous shell. Nevertheless, the β-barrel domain of LSV2 CP shares low sequence identities with other viral CPs.

The P-domain of LSV2 CP contains a long loop with four short β-strands (βE′, βE″, βF′, and βF″) between βE and βF (Fig. 1d). This structure pattern of the P-domain is remarkably longer than those of the $T = 3$ alphanodavirus capsids[24,27,28]. One P-domain extends from the central β-barrel domain and interdigitates with another two P-domains of the homo-trimeric neighboring subunits. The resolutions of all P-domain densities are ~5 Å based on cryo-EM maps (Supplementary Figs. 1–3 and 5–7).

The inner cavity of $T = 4$ LSV2 VLP contains an additional electron-density layer of thickness 3 nm, which is presumably contributed by the randomly distributed NTDs or nucleic acids (Supplementary Fig. 9a). Apparently, the interior surface of the $T = 4$ LSV2 VLP reveals a general electropositive distribution (Supplementary Fig. 9c). Notably, the NTD containing the N-ARM has numerous basic residues, although in a disordered form, which should contribute additional positive electrostatics. The NTDs are presumably located at the center of the homo-trimeric capsomeres and are wedged among the subunits A of the pentameric homo-trimers A/B/C around the icosahedral 5-fold (I5) axes in the $T = 4$ and $T = 3$ capsids (Supplementary Figs. 9b and 11a).

## C-arm of LSV2 CP

The structural model of LSV2 CP terminates at M475 as the last ordered residues of the C-terminus. Upon homo-refinement with the icosahedrally symmetrized reconstruction of $T = 4$ LSV2 VLP, we observed additional but diffused densities beneath the helical domain symmetrically around the I2 and I5 axes. These unclear extra densities possibly from those unassigned main chains make close contacts to engage with the helical domains (Supplementary Fig. 12a). The refinement result indicates that all subunits in the $T = 4$ LSV2 VLP constitute this additional segment. These extra densities from six capsomeres around I2 axes are clearer and more ordered than those from five capsomeres around I5 axes, indicating that these short segments of subunits B/C/D are more rigid than those of subunits A. This condition reflects the divergence of the C-terminus structures and arrangements between subunit A and subunits B, C, or D.

Limited with icosahedrally symmetrized reconstruction with homo-refinement, the unclarified extra densities of the C-terminus around I3 and I5 axes are observed in $T = 3$ LSV2 VLP, consistent with those around I2 and I5 axes in $T = 4$ LSV2 VLP, indicating that the local C-arm distributions deviated from icosahedral symmetry are possible. Masked 3D refinement is a suitable solution to treat the dataset within a single protein self-assembly complex, such as a viral particle, and can permit sub-regions of the viral CP to make small deviations from a strict icosahedral symmetry. We used cryoSPARC's non-uniform focused refinement to improve the resolution of disordered C-terminus regions within the homo-trimeric capsomeres of $T = 4$ and

$T = 3$ LSV2 VLPs on reducing an over-fitting tendency of disordered regions[29].

As a result, the C-arm (residues 477–513) contains 37 amino acids downstream of helix α12 of the C-terminus (Fig. 1c and Supplementary Fig. 12b). The C-arm adopts a hairpin conformation with two loops, which is partially resolved in our reconstructions with a symmetry-imposed mode. Although the C-arms of the subunits A around the I5 axes are more disordered than those of the other subunits (B/C/D), several interactions between the C-arm and the helix α12 result in their rigid and stable conformations. First, residues V479 around the I3 and Q3 axes on $T = 4$ LSV2 VLP mediate the homo-trimeric C-arm/C-arm interactions (Supplementary Fig. 12c). Second, a sequence-based surface analysis indicates that the hydrophobic and van der Waals forces contribute predominantly to the helix α12/C-arm and C-arm/C-arm interactions. Eventually, these C-arms might contribute to the formation and stability of the icosahedral inner shell.

## An anchor loop involved in shell stability

The central β-barrel domain of LSV2 CP comprises a unique anchor loop (residues 356–382) in an extended conformation running across the exterior surface of the neighboring subunit with a clockwise rotation around the corresponding I3 and Q3 axes, respectively (Fig. 2a). In the $T = 4$ capsid, the anchors of subunits A, B, C, and D are positioned on the caps of subunits B, C, A, and D, respectively. We found some close interactions, such as hydrogen bonds, between residues on the central β-barrel domain surface; R360, C362, and E368 located on the βG′–βH′ loop of one subunit interact with D221 on the helix α5, N383 on the βH′–βH loop and R201 on the βC′–βC″ loop of the neighboring subunit to form one homo-trimeric capsomere (Fig. 2b).

Intriguingly, a deletion mutant without the anchor segment of 13 residues (356–378) substantially decreases the yield of intact LSV2 VLPs (Fig. 2c). With size-exclusion chromatography (SEC), we observed that the major proportion of the LSV2 anchor-loop-deletion mutant generally failed to assemble VLPs and remained intermediates that were smaller than the complete VLPs. The fractions of peak 1 and peak 2 were further pooled and isolated with SEC and analyzed with SDS-PAGE and negative-staining EM (Supplementary Fig. 13). As a result, peak 1, containing a mixture of larger CP aggregations, irregular particles, and only a few VLPs, had the propensity for self-interaction without assembly order, whereas the peak 2, with smaller variable intermediates of CP, lost the assembly ability. Compared to full-length LSV2 CPs tending to assemble into complete capsids, the behavior of the deletion mutant confirmed that this identified anchor loop is required for proper viral capsid assembly.

Divalent metal ions, such as $Ca^{2+}$, are typically coordinated with metal-binding residues for the formation, stability, and infectivity of a capsid[30–32]. In the structures of $T = 4$ and $T = 3$ LSV2 VLPs at a resolution of ~2.4 Å, no density was observed for any divalent metal ion other than CPs. The large distances, ~13 Å, between two positively charged residues E260 from each homo-trimeric capsomere positioned along the I3 and Q3 axes indicate that the central β-barrel domains of the LSV2 capsid might be devoid of metal ions (Supplementary Fig. 14). As such, the anchor loops seem to replace the role of metal ions for capsid formation and stability (Fig. 2d).

## Conformational differentiations between LSV2 and delta-N48 LSV1 VLPs

As expected from amino-acid sequence homology, the capsid structure of LSV2 serves as a structural model for those of other closely related stains LSV1 and LSV3. Although LSV2 CP shares 74 and 73% sequence identities with LSV1 and LSV3 CPs, respectively, the major sequence variations are located at the surface-exposed spikes, the anchor loop, and the βB–βC loop on the outer surface. Moreover, LSV1 contains extra 48 residues at the N-terminus (Supplementary Fig. 10).

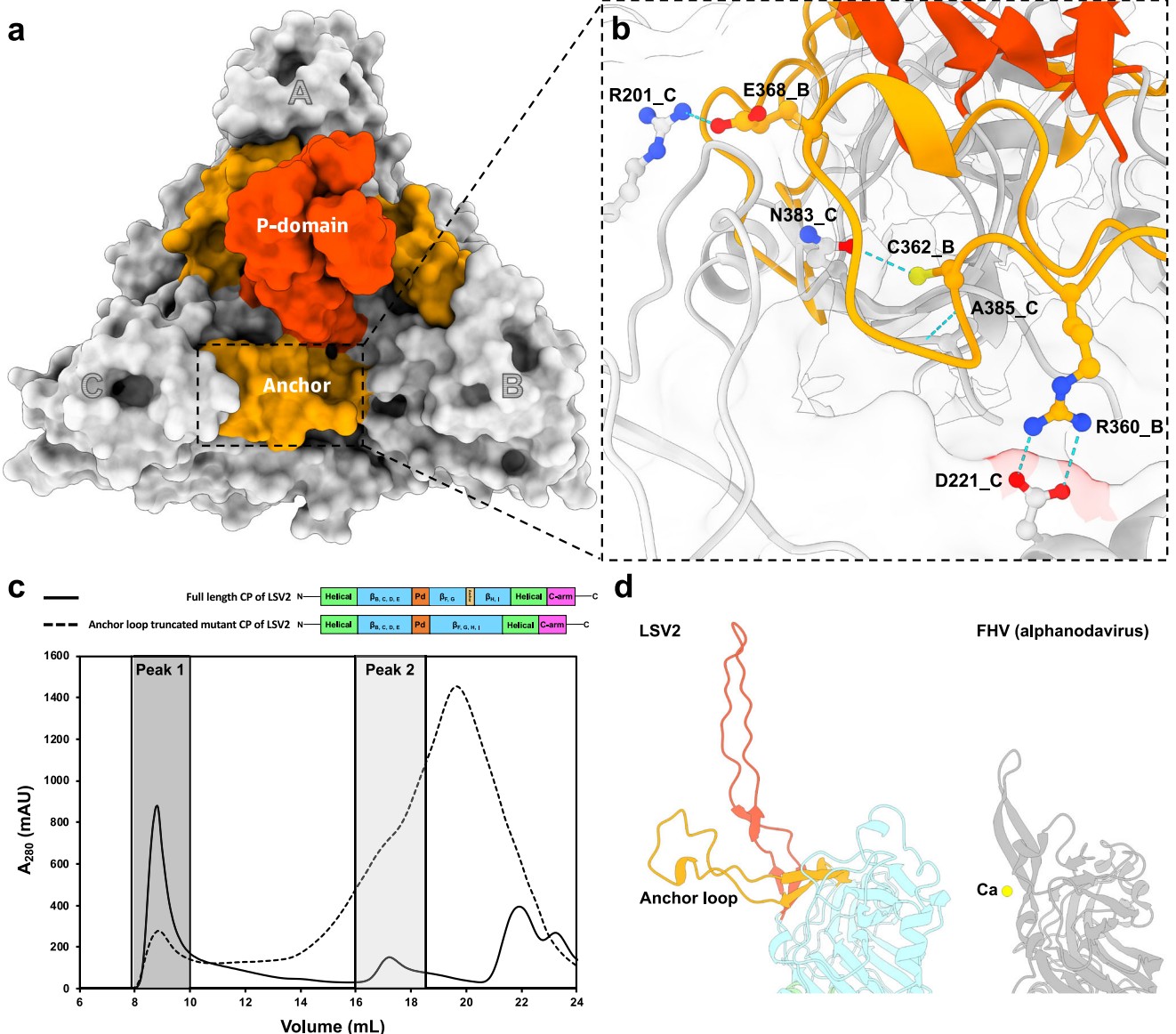

**Fig. 2 | The P-domains and anchor loops form the homo-trimeric capsomere interfaces. a** Surface presentation of one homo-trimeric capsomere on the *T* = 4 LSV2 VLP. The P-domains and anchor loops are colored in salmon red and orange, respectively. **b** The chemical interactions at the interface between two subunits of one homo-trimeric capsomere. The hydrogen bonds are shown with the dotted blue lines. **c** The elution profiles of the full-length (solid line) and the anchor-loop-deletion mutant (dash line) of LSV2 VLPs in the size-exclusion chromatography. The various fractions were collected and their absorbances at 280 nm were measured. The retention volumes of relative proteins contain the major VLPs (such as peak 1) and the single CPs (such as peak 2). **d** The anchor loop of LSV2 CP and one alphanodavirus (FHV) (PDB entry 4FSJ) CP with one $Ca^{2+}$ ion viewed in the same orientation. P-domains and anchor loops are colored salmon red and orange, respectively. The alphanodavirus (FHV) CP and the $Ca^{2+}$ ion are colored gray and yellow, respectively.

We hence explored the residue differences on the outermost surface among LSV1–3[10,18] (Supplementary Fig. 15a).

To investigate the positions of sequence variations in the context of the VLP, we truncated the extra N-terminal 48 residues of LSV1 CP, namely delta-N48 LSV1 CP, which self-assembled in vitro into delta-N48 LSV1 VLP, for cryo-EM structural analysis to inspect the structural differences at the regions of low sequence homology. The delta-N48 LSV1 VLPs were assembled into both *T* = 4 and *T* = 3 particles of diameters ~49 and 45 nm, respectively, similar to the *T* = 4 and *T* = 3 LSV2 VLPs (Fig. 3a). We reconstructed the density maps of the *T* = 4 and *T* = 3 delta-N48 LSV1 VLPs at 4.8 and 3.5 Å resolution, respectively, by gold-standard (0.143) FSC. We could only determine the structure of the *T* = 3 delta-N48 LSV1 VLP because the densities of several large side chains were apparent and the rotamers could be properly modeled. Moreover, some undefined extra densities possibly from unassigned

main chains or nucleotides made close contacts to engage with the inner surface at I5 and particularly I2 axes (Fig. 3a). Finally, we attained the *T* = 3 delta-N48 LSV1 VLP map at a resolution of 2.6 Å after focused refinement and the densities of backbone and side chains were of sufficient quality for model building de novo (Fig. 3b) (Supplementary Table 5).

The cryo-EM structure of the *T* = 3 delta-N48 LSV1 VLP comprises residues 96–558 of subunits A, B, and C, in which residues 49–95 of NTD with N-ARM and the last residues 559–565 of CTD are disordered. The single CP structural model of *T* = 3 delta-N48 LSV1 VLP could be fitted into focused refinement cryo-EM maps at resolutions 4.2 and 2.4 Å of the *T* = 4 delta-N48 LSV1 and LSV2 VLPs with high cross-correlation coefficients of 0.81 and 0.78, respectively, indicating that the single CP structure of *T* = 3 delta-N48 LSV1 VLP greatly resembles those of *T* = 4 delta-N48 LSV1 and *T* = 4 LSV2 VLPs (Fig. 3b, c).

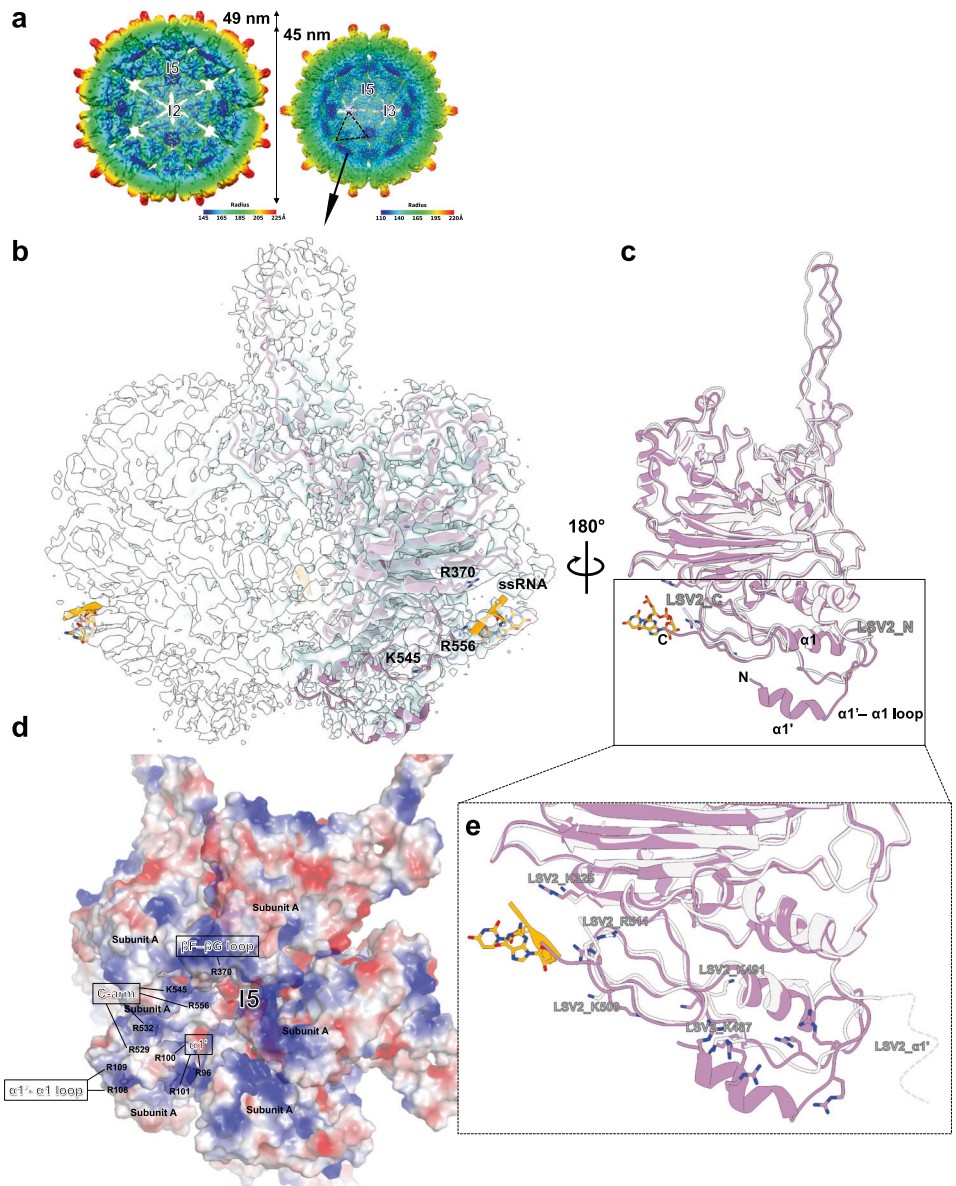

**Fig. 3 | ssRNA binding of _T_ = 3 delta-N48 LSV1 VLP. a** A cutaway view of the _T_ = 4 and _T_ = 3 delta-N48 LSV1 VLPs. A half-sectional view of the _T_ = 4 and _T_ = 3 delta-N48 LSV1 VLPs reveals the obvious extra average density (blue) around the inner surface at I5, I3, and I2 axes. The 3D map is colored by estimates of radius. **b** The portion of the cryo-EM map corresponding to one homo-trimeric capsomere in quasi-equivalent positions of the _T_ = 3 delta-N48 LSV1 VLP. The subunit A is shown as ribbon diagrams colored in purple. ssRNAs are shown in orange. **c** Superposition of the subunits A from _T_ = 3 LSV2 and delta-N48 LSV1 VLPs, and ssRNA shown in silver, purple, and orange, respectively, with a side view relative to Fig. 3b. **d** The electrostatic potential distribution on the inner surface of the _T_ = 3 delta-N48 LSV1 VLP with negatively and positively charged residues colored in red and blue, respectively. The insets show details of the positively charged distribution around the I5 axis. **e** An enlarged view of superimposed subunits A of _T_ = 3 delta-N48 LSV1 VLP with _T_ = 3 LSV2 VLP as Fig. 3c. The key positively charged residues of _T_ = 3 LSV2 VLP are labeled.

Superimposed Cα atoms of LSV2 and delta-N48 LSV1 CPs show a root-mean-square deviation (RMSD) of 0.57 Å.

The helical domain of _T_ = 3 delta-N48 LSV1 CP comprises a three-helix fold as found in LSV2 CP, but is equipped with an additional N-terminal distal helix (α1′, residues 96–104) that associates with the core (α1, α11, and α12) of the helical domain (Fig. 3b and Supplementary Fig. 11b). Importantly, in contrast to global resolution 3.5 Å of the _T_ = 3 delta-N48 LSV1 VLP, the detailed structural features of the helix α1′ and C-arm, as well as α1′–α1 loop particularly, were resolved only with the weak density at the focused refinement at 2.6 Å, indicating that these three regions are more dynamic than the core of the helical domain. Positioned at the inner surface of the capsid, the helix α1′,

α1′–α1 loop, and C-arm of delta-N48 LSV1 CP containing the sort of basic residues (9 arginines) embedded within the inner cavity might have a functional role in viral nucleic-acid encapsidation (Fig. 3c, d). Comparably, the helices α1′ and α1′–α1 loops existing in the _T_ = 4 and _T_ = 3 LSV2 VLPs are invisible due to greater mobility (Fig. 3e).

Not only helix α1′ but also the groove formed between the central β-barrel domain and C-arm possess many positively charged residues for involvement in nucleic-acid encapsidation (Fig. 3d). In addition, the focused refinement reconstruction in _T_ = 3 delta-N48 LSV1 VLP further shows a visualization of the RNA coordination (Fig. 3a, b), in which a two-based single-strand RNA (ssRNA) was partially resolved. K545 and R556 are two basic residues on the C-arm, but only the side chain of

R556 faces the ssRNA-binding site, as does that of R370 (Fig. 3a, b). This ssRNA-binding site is formed by the dynamic C-arm and the βF–βG loop of the central β-barrel domain. The structures and the positively charged residues of the underlying ssRNA-binding regions are almost conserved across the different strains of LSV[10,18], except three positively charged arginine residues (S101 vs. R52, N109 vs. R60, and R522 vs. N484 on LSV1 vs. LSV2, respectively) and the relatively rigid conformations of helix α1′ and the α1′–α1 loop on delta-N48 LSV1 (Fig. 3b–e and Supplementary Fig. 10).

The existence of nucleic acids inside LSV2 and delta-N48 LSV1 VLPs were further analyzed by measuring absorbance at 260 nm ($A_{260}$) and 280 nm ($A_{280}$), respectively. Utilizing these absorbance characteristics, a sum of the components can be calculated from the primary absorbance of the nucleic acid at $A_{260}$ and the CPs at $A_{280}$ after subtracting the background light scattering. The purified LSV2 and delta-N48 LSV1 VLPs showed the $A_{260}/A_{280}$ ratio in the range of 1.68–1.88, respectively, indicating that both VLPs contain nucleic acids (Supplementary Table 8).

Both spikes composed of trimeric P-domains of LSV2 and delta-N48 LSV1 VLPs are flexible with a dynamic clockwise rotation; the P-domains of LSV2 and the delta-N48 LSV1 VLPs are ~4 Å longer than that of $T = 3$ alphanodaviruses[24,27,28] (Figs. 2d and 3c). These facts indicate that an allosteric effect in the P-domain may exist during its dynamic rotation and movement for the receptor binding. Furthermore, both the anchor loops of the LSV2 and delta-N48 LSV1 VLPs are similarly located outside the capsid surfaces, presumably allowing capsid assembly with the conserved key residues for the hydrogen-bond formation (except A413 vs. E368 on LSV1 vs. LSV2), membrane binding and host recognition.

## Mimicking the autoproteolysis cleavage sites for LSV

Previous studies on tetraviruses identified that some residues of two CPs of PrV (D90, A92, G93, E96, T235, Q429, Y439, K507, N556, and F557) and NωV CP (D97, A99, G100, E103, T246, Q443, Y453, K521, N570, and F571) are involved in autoproteolysis cleavage reactions, resulting in γ peptides[23,33,34]. However, all our cryo-EM structures of LSV2 and delta-N48 LSV1 VLPs, which are analyzed not only in a neutral pH condition but also in alkaline (pH 8.5) and acidic (pH 6.5) conditions, have continuous densities for residues 458–463 and 499–505 of LSV2 and delta-N48 LSV1 CPs, respectively, indicating that these two CPs dominantly contain the connected structures from helices α11 to α12. Superimposed structures of central β-barrel domains of LSV2 and delta-N48 LSV1 CPs with PrV and NωV CPs show that the structurally corresponding residues of the hypothetical autoproteolysis site are N81, M83, G84, P85, D86 (S131 on delta-N48 LSV1), T238, Q323, Y331, K411, Y459, N460, and D461 in LSV2, which are different from those of PrV and NωV (Fig. 4a). A CP structural comparison of the $T = 4$ LSV2 and delta-N48 LSV1 VLPs with the tetraviruses and alphanodaviruses shows several interesting facts (Fig. 4b). First, the hydrophilic residue D86 and the uncharged residue S131 are no longer exposed to the cleavage site due to the α1–α2 loop conformation. Second, M83 and N460/D461 in LSV2 CP significantly differ from E96 and N556/F557 and E103 and N570/F571 on the scissile bonds in the PrV and NωV CPs, respectively.

Interestingly, an introduction of a double mutation of M83E/D461F on the LSV2 CP, mimicking the key residues at the autoproteolysis sites of PrV and NωV, potentially triggers a self-cleavage process of LSV2 CP in a neutral pH buffer (Fig. 4c). The double mutant M83E/D461F migrated on SDS-PAGE with a band corresponding to a molecular weight less than that of full-length CP by ~7 kDa (Fig. 4c). Furthermore, we confirmed that each band on SDS-PAGE corresponding to the full-length LSV2 CP and M83E/D461F by in-gel trypsinolysis followed by MALDI-TOF MS-based proteomics (Supplementary Fig. 16). Analyses of the multiple peptides from these two bands result in 79% and 75% sequence coverages for 520 amino acids of LSV2 CP

and M83E/D461F, respectively. As a result, first, two peptides (residues 501–511 and 512–520) consonant with the last 20 residues of the C-terminus were detected only from the full-length LSV2 CP but not from M83E/D461F. Second, an additional peak at 1247.564 m/z was observed only in M83E/D461F and conformed to the specific peptide mass of residues 449–459. The MALDI-TOF MS analyses suggest that M83E/D461F possesses autoproteolysis at the cleavage site Y459–N460 on the scissile bond.

## pH-dependent dynamic particle motions

Cryo-EM is a useful tool to examine pH-dependent structural changes of macromolecules[35]. To investigate available conformational states of the LSV2 VLP, we first analyzed the VLP sizes based on cryo-EM structures in varied pH environments. As a result, the $T = 4$ and $T = 3$ particle sizes of LSV2 VLP are variable, with diameters ranging from 494 to 482 Å and 450 to 438 Å, respectively, from alkaline (pH 8.5) to acidic (pH 6.5) conditions (Fig. 5a, b). A particle comparison of $T = 4$ and $T = 3$ VLPs at varied pHs based on capsid alignment at the icosahedral symmetry shows that all LSV2 VLP particles at basic and neutral pHs have greater expansions than those at acidic pH, particularly the areas at the I2 (Q6) axes on $T = 4$ VLP and I3 axes on $T = 3$ VLP.

In comparison to the expanded $T = 4$ and $T = 3$ particles at the neutral or basic pH condition, measurements of diameters of the shaped holes on the I5 and I2 axes of the unexpanded particles at pH 6.5 show that the minimum and maximum displacements were changed from ~5 to 4 Å and from ~22 to 15.5 Å, respectively (Fig. 5a–c). At a neutral or basic pH environment, homo-trimeric subunits A/B/C rotate along I5 axes; homo-trimeric subunits D/D/D raise around I2 axes, leading the iASU to pivot upward (Fig. 5d). Because of the minimum expanded radius of shaped holes at I5 axes, for the βD–βE loop to maintain the interaction with another four neighboring subunits A at I5 axes, homo-trimeric subunits A/B/C are rotated by further 15°, accompanying with a shift and tilt by 4 Å (Fig. 5d). In contrast, the basis of the quaternary change on the $T = 4$ and $T = 3$ particles at acidic pH is further flattened on the I2 and I3 axes, respectively, suggesting flattened hexameric capsomeres concomitantly result in a compact particle (Fig. 5a–d). Furthermore, the superimposition Cα of CPs between $T = 4$ and $T = 3$ LSV2 VLPs at a neutral (or basic) and acidic pH shows no obvious change.

To gain further structural information about particle sizes in environments of varied pH, we analyzed the size variations of $T = 4$ LSV2 VLP with X-ray crystallography and SAXS. First, the $T = 4$ LSV2 VLP crystals at resolution ~8 Å were grown under varied conditions with pH 6.5–7.0 (Fig. 5e and Supplementary Table 6). The $T = 4$ particle size of a diameter of 450 Å based on the crystal packing is slightly smaller than the observed size range of the cryo-EM $T = 4$ LSV2 VLP, suggesting the potential smallest shaped holes at the I2 (a diameter of 10 Å) and I5 (a diameter of 3.5 Å) axes (Fig. 5a, b). This smallest diameter of the unexpanded LSV2 particle is larger than that of tetraviruses (Supplementary Fig. 17). Notwithstanding this difference of $T = 4$ particle sizes, the crystal and cryo-EM structures of $T = 4$ LSV2 VLP share a similar CP conformation.

Second, for SAXS, the pH environments of LSV2 VLP were directly transformed from pH 7.5 to pH 6.5 and 8.5 in the column, respectively; the full sizes of $T = 4$ and $T = 3$ LSV2 VLPs at pH 6.5 are estimated as ~482 and 438 Å, respectively, whereas those of $T = 4$ and $T = 3$ LSV2 VLPs at pH 8.5 increased to ~494 and 450 Å, respectively (Fig. 5f and Supplementary Table 7). The observed particle conformation discrepancies can be associated with diverse solvent pH conditions and also confirms the cryo-EM results in the cryo-condition. Taken together, structural analyses by cryo-EM, X-ray crystallography, and SAXS indicate that the LSV2 VLP can adjust its capsid conformation according to a varied pH environment, presumably leading to a potential packaging and releasing of the viral genome with a

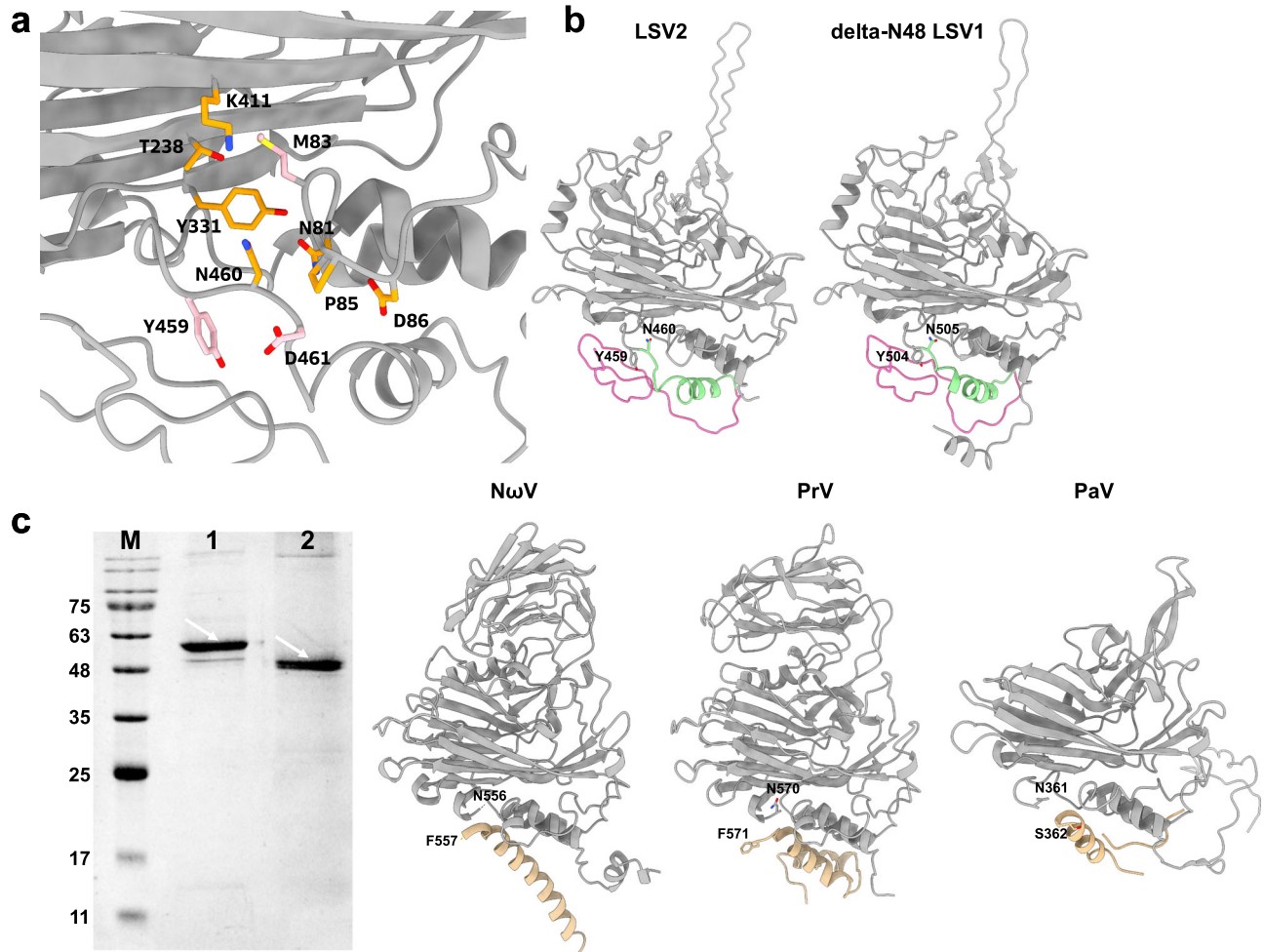

**Fig. 4 | Structure comparison at the autoproteolysis cleavage area. a** The mimicking autoproteolysis site (gray ribbon) in LSV2 CP contains identical key residues (orange sticks) except the key catalytic residues M83, Y459, and D461 (pink sticks), compared to other tetraviruses with γ peptide cleavage. **b** A comparison of scissile bonds among LSV2, delta-N48 LSV1, NωV (tetravirus), PrV (tetravirus), and PaV (alphanodavirus) CPs. The helix α12 and C-arm of the LSV2 and delta-N48 LSV1 CPs are colored in green and pink, respectively. The γ peptides of the NωV, PrV, and PaV CPs are colored in gold. The relevant residues of cleavage sites are shown as sticks. **c** SDS-PAGE analysis of the full-length and mutant type of LSV2 CPs. The full-length LSV2 CP (lane 1) and the double mutant M83E/D461F (lane 2) exhibit the different molecular weights shown with arrows relative to standards (lane M). Five independent SDS-PAGE analyses gave similar results.

conformational transformation between the unexpanded and expanded capsids.

## Discussion

We determine the first LSV VLP capsid structures of two major strains LSV1 and LSV2 of various forms with $T=4$ and $T=3$ architectures with cryo-EM at atomic resolution ~2.3 Å and with X-ray crystallography and SAXS at low resolutions. Both LSV2 and delta-N48 LSV1 VLPs reveal that the single CP retains the assembling ability to form $T=4$ and $T=3$ capsids of multiple sizes simultaneously (Fig. 1a, c). The past study showed that the homo-dimeric capsomeres of Hepatitis B virus (HBV) can assemble into two types of capsids simultaneously, $T=3$ and $T=4$ capsids, but only $T=4$ HBV is the predominant species within infectious virions[36]. Here we provide the full capsid structures of $T=4$ and $T=3$ LSV2 and delta-N48 LSV1 VLPs that reveal several interesting and important points, which are discussed in the following.

### Structural differences among distinct lineages of LSV and other viruses

The CP structures of both full-length LSV2 and delta-N48 LSV1 VLPs can be subdivided into several domains: the flexible NTD; the interior helical domain, adjacent to the viral RNA genome; the central β-barrel domain, at the $T=4$ and $T=3$ capsid shells; the P-domain, extending from the central β-barrel domain to form the surface spikes; the C-arm, involved in the formation and stability of the icosahedral inner shell and the coordination of the viral RNA genome; and the flexible CTD (Fig. 1d). A sequence analysis of CPs among three strains of LSV (LSV1, LSV2, and LSV3) shows that delta-N48 LSV1 shares a high sequence identity, 90%, with LSV3, but LSV2 shares only 74% identity with LSV1 and LSV3. The most variable regions among the three LSV strains are all located on the capsid surface (Supplementary Figs. 10 and 15a), which might reflect consequences in specific host interactions and recognitions, such as the host receptor or membrane binding[37,38]. This observation could potentially correlate with an evolutionary divergence, resulting in the distinct strains of LSV[10,18]. Additionally, the long βB–βC loop, as one variable region of LSV CP, composed of helices α2, α3, and α4, and the strand βB′, bends toward the capsid surface between each capsomere at I2 and I5 axes within the icosahedral lattice, significantly differing from those of the tetraviruses and nodaviruses CPs[23,24,27,28,33,34,39] (Supplementary Fig. 15b).

The conserved key residues of the anchor loop among the different LSV strains, which expose their backbone amide and carbonyl

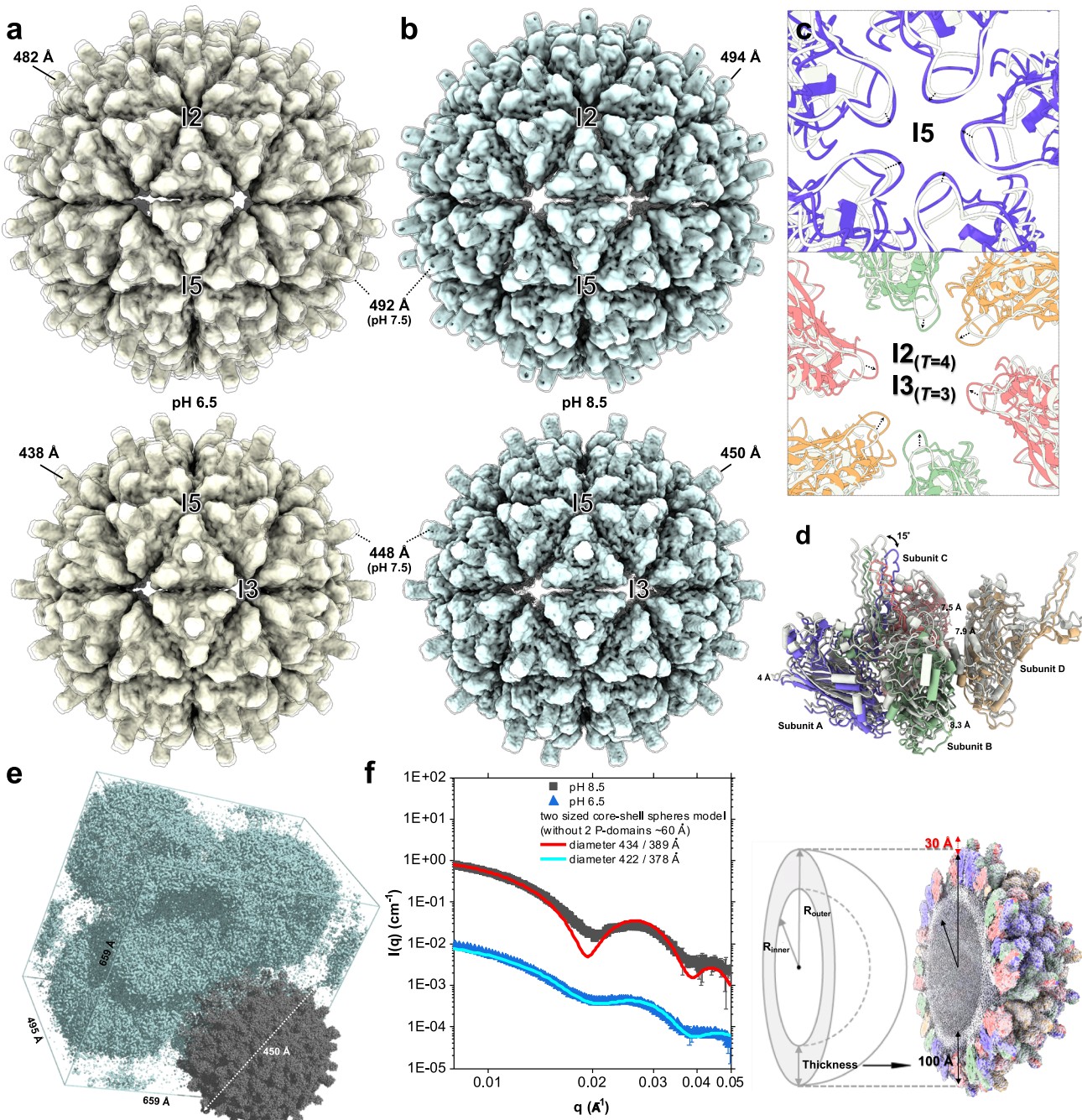

**Fig. 5 | Comparisons of cryo-EM, crystal and SAXS structures of the $T = 4$ and $T = 3$ LSV2 VLPs at neutral, acidic, and basic pH environments. a** The cryo-EM maps of $T = 4$ (upper) and $T = 3$ (lower) LSV2 VLPs with diameters of 482 and 438 Å, respectively, at acidic pH condition are shown in dark khaki, whereas those with diameters of 492 and 448 Å, respectively, at neutral pH condition, are shown in transparent rendering. **b** The cryo-EM maps of $T = 4$ (upper) and $T = 3$ (lower) LSV2 VLPs with diameters of 492 and 448 Å at neutral pH conditions are shown in light blue, whereas those with diameters of 494 and 450 Å at basic pH conditions are colored in transparent rendering. **c** Superposition of the cryo-EM structures of $T = 4$ and $T = 3$ LSV2 VLPs in neutral and acidic environments with top views of the interfaces along I5 (upper) and I2 (lower) axes, respectively. The structures of subunits A–D in an acidic environment are shown in purple, red, green, and yellow,

respectively. All subunits in a neutral environment are shown in silver. **d** A side view of the iASU at neutral and acidic environments. The structures of subunits A–D in an acidic environment are shown in purple, red, green, and yellow, respectively. All subunits in a neutral environment are shown in silver. **e** Molecular packing of $T = 4$ LSV2 VLP in crystals is well arranged in the unit cell of space group $P4_2$. The $2F_o$-$F_c$ electron density is illustrated with green mesh contoured at $1.0\sigma$. The crystal structure of $T = 4$ LSV2 VLP is shown in dark gray. **f** Intensity vs $q$ profile of LSV2 VLPs from SAXS in various buffer solutions. Buffer contribution was subtracted. Two different particle sizes of $T = 4$ and $T = 3$ LSV2 VLPs in pH 8.5 and 6.5 are colored in black and blue, respectively. The spectra represent experimental data and the curves (red and cyan) represent solid sphere models fitting using ATSAS fitting program[61].

groups to form hydrogen bonds with the central β-barrel domain of neighboring subunits, presumably play a crucial role in stabilizing the inter-subunit contacts around the 3-fold axes during assembly (Fig. 2a, b). The mutant with anchor loop deletion of LSV2 CP greatly reduced the VLP formation of the first peak faction in SEC through analyses of SDS-PAGE and negative-staining EM (Fig. 2c and Supplementary Fig. 13); intermediates were predominantly observed in the second peak fraction owing to deficient hydrogen bonds from anchor loops. The anchor loop is thus necessary to perform the formation of the homo-trimeric capsomeres in an earlier stage of LSV capsid assembly.

Our cryo-EM structures of $T = 4$ and $T = 3$ LSV2 and delta-N48 LSV1 VLPs reveal that the extended P-domains of a flexible long-loop conformation, protruding from the central β-barrel domain, are twisted ~25° clockwise (Figs. 2a and 3b). The P-domains of the two major LSV strains show no significant structural differences, despite sharing low sequence identities (Supplementary Figs. 10 and 15a). These two LSV CP structures and sequence identities differ from those of tetraviruses PrV and NωV, which belong to families *Carmotetraviridae* and *Alphatetraviridae*, respectively[16,17,23,34]. Furthermore, the P-domains of LSVs significantly differ from those of the previously reported Ig-like domains of PrV and NωV, and are also distinct from those in $T = 3$ alphanodaviruses in terms of the structural characteristics of length, tilt, and twisted angles (Fig. 4b). Nevertheless, the characteristics of tetraviruses not only have a very restricted host range that is limited to lepidopteran insects but also reflect nonenveloped icosahedral $T = 4$ virions with a diameter of ~42 nm containing either monopartite or bipartite positive-sense ssRNA genomes, similar to LSVs in our study (Supplementary Fig. 17). As above mentioned, we anticipate that LSVs, currently classified into the family *Sinhaliviridae*, belong to the genus of tetraviruses with $T = 4$ virions between the families *Alphatetraviridae*, *Carmotetraviridae*, and *Nodaviridae* that generally infect the larvae of insect species[17,23,24,27,28,33,34,39].

In LSV VLPs, CPs can assemble to homo-trimeric capsomeres without $Ca^{2+}$, compared with tetraviruses and nodaviruses CPs (Supplementary Fig. 14). One conserved negatively charged aspartic acid residue (D281 on LSV2, D326 on LSV1, and D279 on LSV3) located at the top region of the P-domain is identified in all three strains of LSV (Supplementary Fig. 18); the P-domain, as well as the anchor loop, of LSVs might thus result in a compact conformation of the homo-trimeric capsomeres, potentially triggered by an environment with concentrated divalent-metal ions. Negatively charged residues on the viral P-domain are a general characteristic of capsid assembly, such as in the case of betanodaviruses[31]. It is thus worthy to investigate the high-resolution structures of LSV P-domain together with analytical methods, such as multi-angle light scattering coupled with SEC and simulations of molecular dynamics in the future. Furthermore, all $T = 4$ and $T = 3$ particles of LSV2 VLPs in various pH environments from alkaline to acidic conditions show the conserved homo-trimeric capsomere arrangement but with dynamic spike conformations (Fig. 5a, b, f). Such structural flexibility of the viral P-domain could facilitate viral assembly and virus-receptor interactions[31,32,37,38]. The results suggest that the P-domain with dynamic rotation and flexibility remains straight and extended in LSV capsids before a viral infection, whereas it could appropriately adapt an angular conformation upon binding to host cells.

## pH-dependent dynamic particle motion and related function
The functional capsid not only protects the viral genome but also plays an important role in the delivery and release of the viral genome into the host cell. In previous studies, the capsids of several non-enveloped ssRNA viruses initially assemble as an expanded procapsid and then become mature upon a lytic peptide release that is required for the stability and infectivity of an unexpanded particle[39–42]. Although the particle diameters of cryo-EM and SAXS structures of $T = 4$ and $T = 3$

LSV2 VLPs at pH 7.5 and 8.5 are ~10 Å larger than those at pH 6.5, the single CP structures of LSV2 VLPs at pH 6.5, 7.5 and 8.5 and that of delta-N48 LSV1 VLPs at pH 6.5 are conserved (Figs. 4b, 5a–d, f and Supplementary Fig. 11) (Supplementary Table 7). Superimposed Cα atoms of LSV2 and delta-N48 LSV1 CPs show RMSD of 0.57 Å (Fig. 4b and Supplementary Fig. 11). The expending movements of shaped holes at I5 axes observed in all cryo-EM structures of $T = 4$ and $T = 3$ LSV VLPs are less and confined in various pH environments, compared with those at I2 and I3 axes (Fig. 5a–c).

A comparison of the cryo-EM and SAXS structures of $T = 4$ LSV2 VLPs and the crystal structure of $T = 4$ LSV2 VLP show that the dynamic particle sizes are well correlated with the diameters of shaped holes at I2 axes under the various pH environments (Fig. 5a–f and Supplementary Fig. 17). It is important to emphasize that the pore radii of shaped holes at I2 axes on $T = 4$ LSV VLPs are more dynamic and variable than that at I5 axes, suggesting that the shaped holes at I2 axes could function and participate in opened or closed capsid structures and genome releasing in vivo.

## Mimicking the autoproteolysis process with key mutations
The C-termini in LSV2 and delta-N48 LSV1 CPs remain complete, not proteolyzed, and exist in all cryo-EM structures at pH ranging from 6.5 to 8.5. The LSV2 and delta-N48 LSV1 CPs both exhibit the structural flexibility on the NTD and the rigid conformation of the C-terminus, with continuous density until G520 and R558, respectively, folding into one short helix α12 and the C-arm that occupy positions nearby the helical domain around I2, I3, and I5 axes without γ peptide formation (Supplementary Figs. 11 and 12b). In contrast, the conformation and multifunction of γ peptides from C-termini of PrV, NωV, and alphanodavirus CPs are composed of one or two long α-helices, and are responsible for cell membrane lysis, RNA coordination, and capsid stabilization[27,33,34,39] (Fig. 4b). Furthermore, both E103 and N570 mutants of NωV CPs blocked the autoproteolysis cleavage process, resulting in an aberrant particle assembly, even at pH 5.0[40,43,44].

An absence of the autoproteolysis process suggests that the single CP structure remains invariant and all strains of $T = 4$ LSV could adopt at least two reversible capsid sizes (expanded and unexpanded) among varied pH environments, similarly to mutant-type NωV particles (N570T)[40]. Although N505 of LSV1 and N460 of LSV2 CP are consistent with N556 of PrV CP, N570 of NωV CP and N361 of PaV CP on the scissile bonds, M128/D506 and M83/D461 in LSV1 and LSV2 CPs differ significantly from E96/F557 and E103/F571 in PrV and NωV CPs and P67/S362 in PaV CP, respectively (Fig. 4a, b). More interestingly, we observed that the double mutant M83E/D461F in a neutral pH buffer enable LSV2 CP to exhibit the autoproteolytic capability (Fig. 4c). Mass spectrum analysis further showed that the cleavage site is between Y459 and N460 (Supplementary Fig. 16), presumably releasing a C-terminus peptide. This autoproteolytic activity might be achieved when the catalytic M83E attacks the scissile bond (Y459–N460) that is moved structurally closer to the M83E residue while D461 is mutated to Phe. Further investigations are needed to elucidate the mechanism of autoproteolysis cleavage with the associated catalytic residues and the scissile bond. All the above differences in dynamic particle sizes with pH variation and C-termini between the various strains of LSV and tetraviruses might imply that the C-termini of LSV CPs exhibit other functional properties different from the conventional multifunction of the γ peptide. Further investigation is needed to determine the factors that control the efficiency of the quaternary rearrangements that accompany maturation or viral genome release in LSV.

## Viral RNA genome encapsidated by the NTD, helical domain, β-barrel domain, and C-arm
The cryo-EM structural analysis reveals ssRNA molecules bound in the inner space of the $T = 3$ delta-N48 LSV1 VLP, whereas no ssRNA was coordinated in $T = 4$ and $T = 3$ LSV2 VLPs. The $A_{260}/A_{280}$ ratios of the

LSV2 and delta-N48 LSV1 VLPs agree with the expected UV $A_{260}/A_{280}$ ratio of RNA encapsidation by assembled VLPs in vitro of FHV[45]. Several VLPs have been shown to predominantly package host RNA and its encoding RNA[46,47]. Our results suggest that capsids can incorporate *E. coli* RNA which probably interacts with positively charged residues on some RNA-binding regions of LSV CPs (Supplementary Table 8). The additional asymmetric densities on the inner surface of all cryo-EM structures of LSV VLPs might imply the locations, particularly at I5 axes, of the flexible RNA-binding regions and undefined RNA molecules (Fig. 3a and Supplementary Fig. 9a, b). Moreover, the irregularly undefined electron densities of RNA-binding regions, such as the helix α1′ and α1′−α1 loop, seem to be a consequence of the high flexibility of the α1′−α1 loop (Fig. 3a–c). Finally, N-terminal residues 1–48 existing only in LSV1 CP might not participate in RNA encapsidation due to comprising only a few positively charged residues (Supplementary Fig. 10).

Despite overall structural similarities between delta-N48 LSV1 and LSV2 VLPs, a possible explanation for ssRNA observation in only $T = 3$ delta-N48 LSV1 VLP is that the relatively rigid helix α1′ and α1′−α1 loop of delta-N48 LSV1 CP can coordinate the access of ssRNA to the capsid inner surface and subsequently this ssRNA approaches to the interface between the neighboring subunits to allow the additional short ssRNA molecules to be accommodated at I5 and I2 axes (Fig. 3b–e). As a result, first, the helix α1′ with R96, R100 and R101 and α1′−α1 loop with R108 and R109 of delta-N48 LSV1 CP seem to participate as a velcro to trigger the ssRNA attachment near the inner capsid surface, possibly during ssRNA encapsidation. Subsequently, most other positively charged residues required for the RNA-binding ability, including R370 on the βF−βG loop and R529, K532, K536, K545, and R556 on the C-arm of delta-N48 LSV1 CP, are located at the periphery at which they might eventually function to promote RNA contacts at interfaces with neighboring subunits. Consequently, only a short rigid ssRNA is seen to bind with three nearby positively charged residues, i.e., R370, K545, and K546, of $T = 3$ delta-N48 LSV1 CP (Fig. 3d). In contrast, although the conserved positively charged residues K325 on the βF−βG loop and K487, K491, K500 and K511 on the C-arm of LSV2 VLP are also located at the inner surface of LSV2 VLP, a thick electron-density layer are presumably contributed by the potential RNA-binding regions including the highly flexible NTD with the N-ARM motif, helix α1′ and α1′−α1 loop as well as the RNA (Fig. 3e and Supplementary Fig. 9a, b).

## Trapezoid-shaped capsomeres as the basis of VLP architecture

In some cryo-EM images of LSV2 VLP, we observed multiple linear structures composed of several capsomeres of trapezoid shape that form dominos (Supplementary Fig. 19a, b). Each trapezoid-shape capsomere is proposed to provide a basis to build the VLP structure with its shape and size in length 190 Å and width 80 Å consistent with three homo-trimeric capsomeres (Supplementary Fig. 19c). Notably, the anchoring interactions should first occur to integrate the three subunits into one homo-trimeric capsomere to form a structurally rigid unit for assembly. The present evidence described above seems to indicate that the association of nucleic acids with the multiple homo-trimeric capsomeres may serve to trigger the assembly of the trapezoid-shaped capsomeres during the capsid assembly of LSV.

The structural basis of the trapezoid-shaped capsomeres and the domino-scaffold configuration might provide some insight into the mechanism of capsid assembly. Based on these models, we propose the following process of the subunit integration into one complete capsid (Supplementary Fig. 19c). First, the functionality of the anchor loop and P-domain initiates the capsid assembly primarily by modulating the subunit-subunit interactions of the homo-trimeric capsomeres. Second, the nucleic acid can subtly associate with the NTDs, the helical domains, and C-arms of homo-trimeric capsomeres. Third, these interactions further integrate three homo-trimeric capsomeres into the trapezoid-shaped capsomeres that subsequently form a structural domino configuration for capsid assembly. In broad terms, the length of a domino-scaffold configuration could result in $T = 4$ and $T = 3$ shells with a different number of trapezoid-shaped capsomere repeats. On the other hand, some aberrant assemblies were reported to be more easily formed[48,49]. It is thus also possible that trapezoid capsomeres might be off-path or dead-end products during assembly. The observed trapezoid-shaped capsomeres require more characterization in depth in the future.

In summary, the high-resolution cryo-EM structures of LSV2 and delta-N48 LSV1 VLPs not only provide substantial structural insight into the viral capsids but also allow a detailed description of the RNA-binding site and the driving factors for the capsid assembly. Despite the evolutionary and biological divergence of LSV that has been classified into at least eight subgroups[10,18], many gene sequences of CP are highly conserved among various strains of LSV. Our study provides a structural model for all strains of LSV that may self-assemble into common $T = 4$ tetravirus structures with their single CP subunits and infect host cells through a similar entry pathway.

## Methods

### Preparations of LSV2, delta-N48 LSV1 and mutant VLPs

The consensus DNA sequence coding for the LSV1 (GenBank accession no. ASS83276.1) and LSV2 (GenBank accession no. AEH26188.1, https://www.ncbi.nlm.nih.gov/protein/335057591) CPs were synthesized with codons optimized for *E. coli* expression. The sequence coding for the full-length LSV2 CP (encoding residues 1–520) and delta-N48-LSV1 CP (residues 49–565) were amplified by PCR and then subcloned into an artificial modified pET32-Xa/LIC expression vector (Novagen) between the *Sfo*I and *Xho*I restriction sites, to produce N-terminal hexa-histidine-SUMO-tagged fusion proteins[31,32]. The double mutant M83E/D461F and the anchor-loop-deletion mutant of LSV2 CP were prepared using the Quikchange II Site-directed Mutagenesis Kit (Agilent). All proteins were expressed in *E. coli* BL21-CodonPlus(*DE3*)-RIL strain with the addition of isopropyl-β-D-thiogalactopyranoside (IPTG, Bioshop) (final concentration 0.4 mM) at 18 °C overnight. The cells were harvested and resuspended in a binding buffer (50 mM Tris-HCl, pH 8.0, 0.25 M NaCl, 20 mM imidazole, 5 mM β-mercaptoethanol, 1% (v/v) glycerol, 1 mM EGTA, 1 mM phenylmethylsulfonyl fluoride and 20 μg DNase I), and subsequently lysed with sonication. The lysate was centrifuged ($10,000 \times g$) for 25 min; the supernatant was collected and filtered through a filter (0.22 μm), and then injected into a Ni-NTA column (GE Healthcare). The bound CP was washed with 15-column volumes of the washing buffer, which was supplemented with imidazole at a final concentration of 30−50 mM into the binding buffer. The bound CP was eluted with a linear gradient of imidazole (100−500 mM). Finally, the N-terminal SUMO-fusion tag was cleaved off with SUMO protease Ulp1 and was removed through a Ni-NTA column (GE Healthcare). The various purified LSV2 and mutant LSV2 CPs were diluted to a concentration of 0.3 mg/mL and dialyzed against VLP-formation buffers (50 mM HEPES and 300 mM NaCl) at three pH 6.5, 7.5, and 8.5, respectively, at 4 °C overnight; the purified delta-N48 LSV1 CP was dialyzed against another VLP-formation buffer (50 mM Na$_2$HPO$_4$, 50 mM NaHPO$_4$, 500 mM NaCl, 1 mM EDTA, 0.03% Tween-20, pH 6.5) at 4 °C overnight. Finally, the dialyzed CP was further purified on a size-exclusion chromatography with the Superdex 200 Increase 10/300 GL and Superose 6 10/300 GL columns (GE Healthcare) equilibrated in VLP-formation buffers. Fractions containing $T = 4$ and $T = 3$ particles of LSV2 and delta-N48-LSV1 VLPs were concentrated at 30 mg/mL and stored at 4 °C for further cryo-grid preparation and crystallization.

### Negative-staining electron microscopy

The purified and assembled VLP samples were diluted to a final concentration of 50 μg/mL and blotted on the freshly glow-discharged, carbon-coated 200 mesh copper grids (EMS, USA) for 1 min. The filter

paper was used to blot away an excess sample. The sample on the grid was subsequently washed once with ddH$_2$O followed by blotting. Grids were negatively stained with uranyl acetate solution (5 µL, 2% w/v) for imaging with JEM1400 at 120 keV and recorded by a Gatan Ultrascan 4000 CCD-camera model 895 (4k × 4k) (Gatan Inc., USA) at magnification 60,000×.

## Cryo-electron microscopy imaging

Samples for Cryo-EM were prepared using a Thermo Scientific Vitrobot (Mark IV), which was set to 100% humidity at 4 °C. An aliquot (~3.5 µL) of purified VLP solution (30 mg/mL) was applied to the glow-discharged Quantifoil R1.2/1.3 holey carbon grid (Quatifoil GmbH, Germany). After standing for 10 s, grids were blotted with filter paper for 3.5 s with blot force 5 and rapidly plunged into liquid-nitrogen precooled liquid ethane for vitrification. The cryo-EM grids were stored in liquid nitrogen before imaging.

Cryo-EM grids were first verified on a 200 keV Talos Arctica transmission electron microscope (Thermo Fisher Scientific) equipped with a Falcon III detector (Thermo Fisher Scientific) operated in a linear mode. Images were recorded at nominal magnification 92,000×, corresponding to pixel size 1.1 Å/pixel; the defocus was set to −3.0 µm. The identified suitable cryo-EM grid was recovered and stored in liquid nitrogen until data collection by Titan Krios (Thermo Fisher Scientific) at the Academia Sinica Cryo-EM Center. Data sets were automatically collected with EPU-2.7.0 software (Thermo Fisher Scientific) on a 300 keV Titan Krios™ (Thermo Fisher Scientific) equipped with X-FEG electron source and K3 Summit detector (with GIF Bio-Quantum Energy Filters, Gatan) operated in super-resolution mode (gun lens 4, spot size 5, C2 aperture 50 µm). The raw movie stacks were recorded at nominal magnification 105,000×, corresponding to pixel size 0.415 Å/pixel. The defocus range was set to −1.5 to −2.0 µm; the slit width of energy filters was set to 20 eV. For data collection of LSV2 VLP at pH 6.5 and 8.5, 40 frames of non-gain normalized tiff stacks were recorded with a dose rate ~16 e$^-$/Å$^2$ per s; the total exposure duration was set to 2.5 s, resulting in an accumulated dose ~40 e$^-$/Å$^2$ (~1 e$^-$/Å$^2$ per frame). For data collection of LSV2 VLP at pH 7.5 and delta-N48 LSV1 VLP at pH 6.5, 50 frames of non-gain normalized tiff stacks were recorded with a dose rate ~20 e$^-$/Å$^2$ per s; the total exposure time was set to 2.5 s, resulting in accumulated dose ~50 e$^-$/Å$^2$ (~1 e$^-$/Å$^2$ per frame). The parameters for cryo-EM imaging were summarized in Supplementary Table 1.

For imaging the trapezoid-shaped capsomeres and the domino-scaffold configuration, the purified LSV2 VLP sample (3.5 µL, 30 mg/mL) was applied to glow-discharged Quantifoil R1.2/1.3 holey carbon 200-mesh copper grids (Quantifoil GmbH, Germany). Grids were blotted for 4–5 s at 100% humidity and flash frozen with liquid-nitrogen-cooled liquid ethane using a Leica EM GP2 (Leica). The grid was then loaded onto FEI Titan Krios™ EM operated at an accelerating voltage of 300 keV at the Stanford-SLAC Cryo-EM Center. Image stacks were recorded on a Gatan K3 Summit direct detector (Gatan) set on a super-resolution counting mode using SerialEM[50], with a defocus range between −0.8 and −1.4 µm. The total exposure duration was set to 3 s with each subframe duration 0.05 s for 60 subframes per image stack. The total electron dose was 50 e$^-$/Å$^2$ (~0.83 e$^-$/Å$^2$ per subframe). The movies (1.096 Å/pixel) were subsequently aligned and summed using the MotionCor2[51] software to obtain a final dose-weighted image. The contrast transfer function (CTF) was estimated using the CTFFIND4[52].

## Data processing

Super-resolution image stacks were motion-corrected and dose-weighted using MotionCor2[51] with a patch of 5 ×5 and a two-fold binning (resulting in pixel size 0.83 Å/pixel). CTF information was estimated from the images after motion correction and dose weighting

with CTFFIND4.1[52]. Semi-automated reference-free particle picking was performed with cisTEM[53]. The coordinates of the selected particles were imported into RELION 3.0[54] for particle extraction with a box size of 650 pixels. Multiple rounds of 2D classification were performed in RELION 3.0[54] follow by particle selection and extraction. In the final round of 2D classification, the particles of $T = 4$ and $T = 3$ VLPs were separated and re-scaled with a box size of 512 pixels, resulting in a pixel size of 1.0537 Å/pixel. The models of $T = 4$ and $T = 3$ VLPs were calculated ab initio with cisTEM[53] and then imported into RELION 3.0[54] for further 3D auto-refine with icosahedral symmetry (I1). After CTF refinement and Bayesian polishing, the polished shiny particles were imported into cryoSPARC[29] for further 2D classification and 3D heterogeneous refinement without imposing symmetry (C1). The particles belonging to the effective 3D classes were then refined to improved resolution by homogeneous refinement with icosahedral symmetry (I).

To improve the resolution of the CP subunits, we performed a focused refinement[55] to refine further the densities of A/B/C trimers and D/D/D trimers in the $T = 4$ and $T = 3$ VLPs, respectively. First, the symmetry expansion and the local refinement with a soft mask focus on the A/B/C or D/D/D trimer without imposing symmetry (C1) were sequentially undertaken. The features of the focus-refined maps were improved, compared with that of the standard icosahedral reconstructed maps, with a resolution better than 3 Å, which is sufficient for accurate model building de novo. Map sharpening and resolution estimation were made effective in cryoSPARC[29]. The overall resolution was estimated with a criterion Fourier Shell Correlation (FSC) = 0.143; the local resolution was also calculated in cryoSPARC[29]. The 3D density maps were visualized in UCSF Chimera[56]. The procedures of data processing were summarized in Supplementary Figs. 1–4. The details of cryo-EM reconstruction were summarized in Supplementary Figs. 5–8. The statistics of icosahedral reconstructions are summarized in Supplementary Table 1.

## Atomic model building and refinement

The initial atomic models of the $T = 4$ LSV2 VLP at pH 7.5 and $T = 3$ delta-N48 LSV1 VLP at pH 6.5 were built de novo into the focused refinement densities of A/B/C trimers using Coot[57]. Except for N- and C-termini segments, the polypeptide chains of the homo-trimeric subunits A/B/C for residues 66–520 of $T = 4$ LSV2 CP and residues 96–565 of $T = 3$ delta-N48 LSV1 CP were traced. The homo-trimeric subunits A/B/C of $T = 4$ LSV2 VLP at pH 7.5 served as templates for the homology model building of the homo-trimeric subunits D/D/D of $T = 4$ LSV2 VLPs at pHs 6.5, 7.5, and 8.5, and the homo-trimeric subunits A/B/C of $T = 3$ LSV2 VLPs at pHs 6.5, 7.5, and 8.5. Subsequently, each subunit domain was manually fitted into the maps with UCSF Chimera[56] and further manually adjusted and corrected in Coot[57]. The statistics of model validations refer to the final output files from the real space refinement in PHENIX[58]. All statistics of data collection and model refinement are presented in Supplementary Tables 2–5.

## Protein crystallization

Crystallization screening was first performed with the hanging-drop vapor-diffusion method by a liquid-handling robot (Mosquito, TTP Labtech). Preliminary crystals of LSV2 VLP were obtained by mixing VLP (20 µL, 30 mg/mL) with a reservoir solution (20 µL) containing MPD (25–35%, v/v), magnesium acetate (40 mM) and MES (0.1 M, pH 6.5–7.0) at 18 °C. This condition was further optimized to improve the diffraction quality and resolution of the crystals. Crystals of rod shape appeared within 1–2 days and grew to a maximum size of ~0.2 mm in 1–2 weeks. The crystals were transferred from a crystallization drop into a cryoprotectant solution (2 µL) with a reservoir solution containing glycerol (25–30%, v/v) for a few seconds, mounted on a synthetic nylon loop (0.1–0.2 mm; Hampton Research) and then flash-cooled in liquid nitrogen before data collection.

### X-ray data collection, structure determination, and refinement

The LSV2 VLP crystals were simultaneously examined with X-ray diffraction at beamlines TLS 15A1 and TPS 05A1 of the National Synchrotron Radiation Research Center (NSRRC, Hsinchu, Taiwan) during crystallization optimization. The complete data set of LSV2 VLP crystals was collected at beamline BL44XU at SPring-8 (Harima, Japan) with a CCD detector (MX300-HE) with an X-ray wavelength of 0.9 Å. X-ray diffraction data to 8 Å resolution were collected at 100 K. The data were indexed, integrated, and scaled with HKL2000[59]. Details of data statistics are shown in Supplementary Table 6. The $T = 4$ LSV2 VLP crystals belong to the tetragonal space group $P4_2$ with unit-cell dimensions $a = b = 659.96$ Å, $c = 495.74$ Å, and contain one viral particle in an asymmetric unit resulting in 60-fold NCS. The structure of $T = 4$ LSV2 VLP was determined with molecular replacement (MR) using our previously determined cryo-EM structure of $T = 4$ LSV2 VLP as the search model with the program Phaser[58]. One clear MR solution is shown, in which one particle is located at (1/4, 1/4, 0) in one unit cell. All graphics for the molecular structures were generated with UCSF Chimera[56], ChimeraX[60], and PyMOL (The PyMOL Molecular Graphics System, Version 1.2r3pre, Schrödinger, LLC.)

### Scanning SAXS/HPLC and data analysis

LSV2 VLPs were measured at beamline TPS 13A BioSAXS, using the integrated system of the size-exclusion high-performance liquid-chromatography (Agilent chromatographic system 1260 series) and SAXS/UV–Vis/RI detection at NSRRC. Sample solutions were injected into the size-exclusion column with an appropriate flow rate, and directed through a quartz capillary (2.0 mm dia.) thermostated at 283 K for simultaneous SAXS and UV–Vis absorption measurements at the same sample position (with orthogonal incidences). With X-ray 15 keV and two sample-detector distances of 5,000 and 7,044 mm, the data covered $q$-ranges 0.0041–0.2 Å$^{-1}$ and 0.0026–0.2 Å$^{-1}$. SAXS data were collected successively during HPLC sample elution with one data frame per 2 s using Eiger X 9 M in the vacuum detector. Buffer solutions (50 mM HEPES and 300 mM NaCl, pH 6.5 and pH 8.5, respectively) were measured before sample signals. Data were corrected for electronic noise, sample transmission, and detector sensitivity. The 2D SAXS patterns were radially averaged and scaled to the absolute intensity (i.e., scattering cross-section per unit volume) via the absolute water scattering intensity I(0) = 0.01662 cm$^{-1}$ at the same sample temperature. 1D SAXS profiles were evaluated for radiation damage, background-subtraction quality, and sample concentration effects; well-overlapped SAXS profiles collected over the sample elution signal of HPLC were integrated for improved data statistics. The acidic sample solution (1 mg/mL, 55 μL, pH 6.5) was injected into the Tosoh G6000 pwxl column with a flow rate of 0.2 mL/min. The basic sample solution (1 mg/mL, 100 μL, pH 8.5) was injected into the Agilent Bio SEC-3 column with a flow rate of 0.35 mL/min. SAXS data evaluation was performed with the ATSAS package[61]. SASView 5.0.4 was used to fit the SAXS data to the two-sized spherical core-shell model[51]. For LSV2 VLPs in solution from pH 7.5 to 6.5, SAXS data at pH 6.5 was fitted with two core-shell sphere models (with $R_{core1} = 108.7$ Å, $R_{core2} = 86.5$ Å, and $t_{Shell} = 102.3$ Å) and some aggregate interference. For LSV2 VLPs in solution from pH 7.5 to 8.5, SAXS data at pH 8.5 was fitted with two core-shell sphere models (with $R_{core1} = 114.8$ Å, $R_{core2} = 92.1$ Å, and $t_{Shell} = 102.3$ Å). We used simple core-shell sphere models without considering the arch-like structure of the P-domain (length ~30 Å).

### In-gel trypsin digestion and MS analysis

The major CP protein bands corresponding to the full-length LSV2 and double mutant M83E/D461F on SDS-PAGE were manually excised and subsequently cut into small pieces (~0.5 mm³). The excised gel pieces were washed with the solution containing $NH_4HCO_3$ (25 mM) and methanol (40%) followed by acetonitrile (100%) washing. The proteins in gel pieces were treated with DTT (10 mM) and then with iodoacetamide (55 mM). CP Proteins were digested with trypsin (Promega) to give a final ratio of substrate: trypsin at 50:1 in a buffer containing ammonium bicarbonate (25 mM) and acetonitrile (10%, v/v) for 12–16 h at 37 °C. The reaction was stopped by adding formic acid (5%).

An aliquot (0.5 μL) of the supernatant from the digestion was deposited onto the 384/600-μm MTP AnchorChip (Bruker Daltonik GmbH) and allowed to be air dried at room temperature. Subsequently, an aliquot (0.5 μL) of CHCA (1.4 mg/mL α-Cyano-4-hydroxycinnamic acid in buffer containing 0.1% trifluoroacetic acid, 1 mM ammonium phosphate, and 85% acetonitrile) was added and allowed to be air dried. The MALDI-TOF MS analysis was performed in a positive ion mode with delayed extraction (reflection mode) on a Bruker Autoflex maX MALDI TOF/TOF mass spectrometer (Bremen, Germany) equipped with a 200 Hz SmartBean Laser. Data acquisition and processing were carried out manually using FlexControl 3.4 and FlexAnalysis 3.4 (Bruker Daltonik GmbH), respectively. The processed data were further analyzed with Biotools 3.2 (Bruker) packaged by accessing the Mascot server online (https://www.matrixscience.com/) to identify the corresponding polypeptides.

### UV spectroscopy

The existence of the RNA content of the LSV2 and delta-N48 LSV1 VLPs was examined by measuring the absorbance at UV wavelengths of 260 nm ($A_{260}$) and 280 nm ($A_{280}$) with NanoVue Plus UV–Vis Spectrophotometer (GE Healthcare). A matching formulation buffer blank was used for zero adjustments of the instrument before the measurements of each sample. The UV $A_{260}/A_{280}$ ratios were calculated with the corrected $A_{260}$ divided by the corrected $A_{280}$.

### Reporting summary

Further information on research design is available in the Nature Portfolio Reporting Summary linked to this article.

## Data availability

Cryo-EM maps generated in this study have been deposited to the Electron Microscopy Data Bank (EMDB) under accession codes EMD-33190 ($T = 4$ LSV2 VLP at pH 7.5, global refinement), EMD-33368 ($T = 3$ LSV2 VLP at pH 7.5, global refinement), EMD-33369 ($T = 4$ LSV2 VLP at pH 6.5, global refinement), EMD-33370 ($T = 3$ LSV2 VLP at pH 6.5, global refinement), EMD-33371 ($T = 4$ LSV2 VLP at pH 8.5, global refinement), EMD-33372 ($T = 3$ LSV2 VLP at pH 8.5, global refinement), EMD-33384 ($T = 4$ delta-N48 LSV1 VLP at pH 6.5, global refinement), EMD-33373 ($T = 3$ delta-N48 LSV1 VLP at pH 6.5, global refinement), EMD-33374 (A/B/C trimer of $T = 4$ LSV2 VLP at pH 7.5, focused refinement), EMD-33375 (D/D/D trimer of $T = 4$ LSV2 VLP at pH 7.5, focused refinement), EMD-33376 (A/B/C trimer of $T = 3$ LSV2 VLP at pH 7.5, focused refinement), EMD-33377 (A/B/C trimer of $T = 4$ LSV2 VLP at pH 6.5, focused refinement), EMD-33378 (D/D/D trimer of $T = 4$ LSV2 VLP at pH 6.5, focused refinement), EMD-33379 (A/B/C trimer of $T = 3$ LSV2 VLP at pH 6.5, focused refinement), EMD-33380 (A/B/C trimer of $T = 4$ LSV2 VLP at pH 8.5, focused refinement), EMD-33381 (D/D/D trimer of $T = 4$ LSV2 VLP at pH 8.5, focused refinement), EMD-33382 (A/B/C trimer of $T = 3$ LSV2 VLP at pH 8.5, focused refinement), EMD-33383 (A/B/C trimer of $T = 3$ delta-N48 LSV1 VLP at pH 6.5, focused refinement). Their corresponding atomic models have been deposited to the RCSB Protein Data Bank (PDB) under accession numbers 7XGZ, 7XPA, 7XPB, 7XPD, 7XPE, 7XPF, and 7XPG, respectively. The details of all data are present in the Supplementary information. Source data are provided with this paper.

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

## Acknowledgements

We are indebted to Yuch-Cheng Jean and the staff at beamlines TPS 05A, TPS 07A, TPS13A, and TLS 15A1 of the National Synchrotron Radiation Research Center (NSRRC) in Taiwan, and Atsushi Nakagawa, Eiki Yamashita, and the staff at BL44XU of SPring-8 in Japan for beamtime and technical assistance. The cryo-EM experiments of high-resolution structures were performed at the Academia Sinica Cryo-EM Center (ASCEM, supported by Taiwan Protein Project and AS Grant number AS-CFII-108-110) and the cryo-EM data were processed at the Academia Sinica Grid-computing Center (ASGC, supported by Academia Sinica). We especially thank Academician Ming-Daw Tsai for kindly providing the cryo-EM facilities at ASCEM for this study with Taiwan Protein Project (Grant number AS-KPQ-105-TPP and AS-KPQ-109-TPP2). We are grateful to Shang-Rung Wu for the assistance with the negative-staining EM measurement at IMANI of National Cheng Kung University. We thank the Cryo-EM facilities at Stanford-SLAC Cryo-EM Center for imaging. We are indebted to Yung-Hsuan Wu for the technical assistance with the MALDI-TOF/TOF mass spectrometry at the Biophysics Core Facility of the Institute of Molecular Biology and additional data analyses by GRC Mass Core Facility of Genomics Research Center, Academia Sinica. This work was supported by National Science and Technology Council (NSTC) grants 105-2311-B-213-001-MY3, 107-2923-B-213-001-MY3 and 108-2311-B-213-001-MY3, 111-2311-B-213-001, and NSRRC grants to C.-J. Chen.

## Author contributions

C.J.C. initiated the project; C.J.C. and N.C.C. designed the research; N.C.C. designed, expressed, purified viral proteins; C.H.W., N.C.C., M.C.H, S.W. C.J.C. performed cryo-EM imaging and structural determination, N.C.C., M.Y., H.H.G, P.C., C.C.L., P.J.L., Y.C.H., C.J.C. performed X-ray diffraction and analyses; N.C.C., M.Y., C.J.C. determined and refined crystal structures; N.C.C., Y.Q.Y, C.J.C. performed SAXS analysis; N.C.C., C.J.C. performed mutagenesis, UV spectroscopy, and functional assay; N.C.C., M.C.H., C.J.C. analyzed mass spectrometry; N.C.C., C.J.C. analyzed structures; N.C.C., C.H.W., C.J.C. wrote the manuscript and prepared the figures and tables.

## Competing interests

The authors declare no competing interests.
