## [Peer Review File · Nature Communications]

Structures of honeybee-infecting Lake Sinai virus reveal domain functions and capsid assembly with dynamic motionsReviewer #1 (Remarks to the Author):

This manuscript describes the atomic structures of virus-like particles (VLPs) of Lake Sinai virus (LSV) determined using cryo-EM and crystallography. LSVs, taxonomically grouped as belonging to Sinaivirus genus, are divided into 3 (perhaps 4) clades (LS1, LSV2, LSV3). These are recently discovered viruses in honeybees, the ubiquitous pollinators of various vital crops. With high annual losses of honeybee colonies, there has been increased interest in identifying viral (and microbial) pathogens that infect honeybees, and LSV is one among them. Although it is yet unclear whether LSVs are just by-standers or causative agents of loss of honeybee colonies, they are considered targets of concern. In this manuscript, by determining the cryo-EM structures of LSV2 and LSV1 VLPs, which form icosahedral capsids with T=3 and T=4 symmetries, the authors provide the first high-resolution description of the LSV capsid architecture. By determining the structures of LSV in different pH conditions, the authors reveal several interesting structural features of the LSV capsids with implications for capsid stability and assembly. These include: 1) identification of the anchor loop that may function like divalent-metal coordination for capsid assembly, 2) how pH affects the structure and the diameter of VLPs, 3) the possible role of NTD in RNA genome packaging, and 4) the identification of trapezoid-shaped capsomere precursors as possible building blocks for VLP assembly.

The cryo-EM/crystallography analysis is excellent, and the structural observations are very interesting. Considering that LSVs are only recently discovered, the work presents a vital groundwork that may prompt future functional studies to understand the roles of pH-induced particle dynamics, proteolysis of the C-termini, and RNA recruitment during virion assembly for LSV. However, there are some concerns:

1) The manuscript is exclusively structure-oriented, lacking any biochemical studies to correlate structural observations and substantiate the interpretations, which often look very speculative.

2) The interpretation that 'trapezoid' capsomers are the 'domino-configured' building blocks, of the capsid assembly based on the observation of stringed features in cryo-EM images is highly speculative without further experimental evidence. It is quite possible that these features seen in the EM images have nothing to do with capsid assembly and could be dead-end products. Perhaps authors can include this figure in the supplementary material and suggest that these may be building blocks in the discussion.

3) Similarly, the role of NTD in RNA packaging by observing only two bases in the structure of N-terminal deleted VLPs with T=3 symmetry is also highly speculative and unnecessary. Where did the RNA come from in the VLPs? Have the authors checked spectroscopically that these VLPs and not others contain RNA? Even if present, it has to be non-specific interaction and may not have anything to do with capsid assembly or packaging.

4) The crystal structure (at ~8 Å resolution) and the SAXS data do not add much to the manuscript. The changes in the LSV cryo-EM structures at different pHs are already quite convincing.

5) The statement in one of the sentences in the introduction, "presumably belonging to betatetraviruses of the Tetraviridae family with characteristic T=4 quasi-equivalence nonenveloped capsids," is not well supported and perhaps misleading. It is yet unclear whether native infectious LSV viruses have T=3 capsid symmetry or T=4 symmetry. Either the authors should include a supporting citation or remove this part of the sentence.

6) In the sentence "The models of T=4 and T=3 LSV2 VLPs were built and refined with the homo-trimer A/B/C or D/D/D in one asymmetric unit, which form oligomerizes into a homo-tetrameric and a homo-trimeric capsomeres", the part "which form oligomerizes into homo-tetrameric and homo-trimeric capsomeres" is speculative and unnecessary in

this sentence. The authors later suggest that the capsid subunits form homo-trimeric or homotetrameric capsomere. Still, it is unnecessary and somewhat confusing to include it as a part of the above sentence.

7) The description of the structural features in several places can be succinct and less confusing.

8) Overall, the manuscript can be substantially improved by 1) restricting the text to the description of structural features and avoiding unnecessary speculations, 2) revising the subtitles (some of which are redundant) to reflect more appropriately the contents that follow, and 3) removing repetitions of the same idea that occur in several places.

9) "Long spikes (length 3.6 nm) protrude from the center of each trimeric capsomeres at icosahedral 3-fold (I3) or Q3 axes" Should it be 3-fold (I3) and Q3 axes? In the T=4 structure, the spikes are located at both icosahedral and quasi 3-fold axes, whereas in the T=3 structures, they are located at the quasi 3-fold axes.

1) Several typos noted

Reviewer #2 (Remarks to the Author):

In their manuscript, Chen et al. present several viral like particles (VLP) structures formed by the capsid protein (CP) of Honeybee Lake Sinai Virus (LSV1 and LSV2). These viruses have a single-stranded positive sense RNA genome and cause colony collapse disorder in apiaries and have a T=4 icosahedral capsid.

The authors expressed the CP proteins of LSV1 (residues 49-565) and LSV2 (residues 1-520 i.e. the full length protein) in *E. Coli*. Upon dialysis against VLP-forming buffers (at pH 6.5, 7.5 and 8.5), purified CPs self assemble into VLPs with T=3 or T=4 icosahedral symmetry.

The authors determine the structure of T=4 and T=3 LSV2 VLPs at pH 7.5, 8.5 and 6.5 and T=4 and T=3 delta-N48 LSV1 VLPs at pH 6.5, using cryo-electron microscopy (cryo-EM) combined with X-ray crystallography.

They present a detailed description of LSP2 assemblies and compare them with T=4 Nudaurelia capensis ω virus, T=3 pariacoto virus and T=1 infectious bursal disease virus assemblies, which all also have a central beta-barrel domain. They identify an anchoring loop, involved in both the stability of T3 and T4 VLP, while N ω V, PrV and PaV ensure the stability of their capsid with a calcium ion located at the same place as LSV loop.

Previous studies have demonstrated that PrV CP and N ω V CP undergo autoproteolytic cleavage. The authors provide evidence that it is not the case for LSV. However, a double mutation (M83E/D461F) introduced into the LSV2 CP, restoring the autoproteolysis sites of PrV and N ω V, seems to trigger autocleavage of LSV2 CP.

In the refined reconstruction of LSV1 VLP, they observe a two-based single-strand RNA indicative of a genome binding site.

Finally they also observe a compaction of the capsid at low pH.

While I acknowledge the mass of data provided, I am not sure this detailed work will be of interest to researchers beyond the concerned community. As examples, structures of Nudaurelia capensis omega Virus were published in *J. Virol* in 2002, in *Virology* in 2004, in *Protein Science* in 2005, etc.

In any case, I have some comments that should be taken into account.

Major points:

1) In their analysis of the profile of the size exclusion chromatography presented in figure 2C (p7, line 6), the authors mention that the deletion of the anchor segment of 13 residues induces a substantial decrease of the amount of LSV2 VLPs. However they do not comment the broad peak of proteins eluting between ~15 and ~22 ml which exhibits several shoulders. It is unclear why they attribute one of these shoulders more than the

other to monomeric CP. An SDS PAGE gel of the sample before size exclusion would be useful (to see the purity of the proteins) as well as an analysis of the different fractions by SDS PAGE and negative staining.

2) Similarly, the gel presented in figure 3C only suggests that there is a cleavage. The precise site could be analyzed by mass spectrometry. Incidentally, one can notice that the CP proteins although the majority in the gel, are not so pure which reinforces the need of a more in-depth analysis in my previous remark.

3) The authors suggest that the trapezoid-shape capsomeres that they observe in figure 6A constitute the basis of VLP architecture. They may be right but these objects may also correspond to off-path (and even dead-end path) assemblies. Particularly, out of the infectious context, it is known that some aberrant assemblies are more easily formed (e.g. Chevalier et al. *J Virol.* 2004. 78: 3296-303. doi: 10.1128/jvi.78.7.3296-3303.2004).

Minor points:

1) The authors should correct the presentation of the taxonomy in the introduction: the Tetraviridae family has been dissolved in 2011 and replaced by three families

(Alphatetraviridae, Carmotetraviridae and Permutotetraviridae). The genera Omegatetravirus, and Betatetravirus belong to the Alphatetraviridae family.

2) PrV should be identified as Providence Virus somewhere in the text.

3) Several typos and misspelled word in the text which should be corrected.

POINT-BY-POINT responses to the comments of the Reviewers

Reviewer #1

This manuscript describes the atomic structures of virus-like particles (VLPs) of Lake Sinai virus (LSV) determined using cryo-EM and crystallography. LSVs, taxonomically grouped as belonging to Sinaivirus genus, are divided into 3 (perhaps 4) clades (LS1, LSV2, LSV3). These are recently discovered viruses in honeybees, the ubiquitous pollinators of various vital crops. With high annual losses of honeybee colonies, there has been increased interest in identifying viral (and microbial) pathogens that infect honeybees, and LSV is one among them. Although it is yet unclear whether LSVs are just by-standers or causative agents of loss of honeybee colonies, they are considered targets of concern. In this manuscript, by determining the cryo-EM structures of LSV2 and LSV1 VLPs, which form icosahedral capsids with T=3 and T=4 symmetries, the authors provide the first high-resolution description of the LSV capsid architecture. By determining the structures of LSV in different pH conditions, the authors reveal several interesting structural features of the LSV capsids with implications for capsid stability and assembly. These include: 1) identification of the anchor loop that may function like divalent-metal coordination for capsid assembly, 2) how pH affects the structure and the diameter of VLPs, 3) the possible role of NTD in RNA genome packaging, and 4) the identification of trapezoid-shaped capsomere precursors as possible building blocks for VLP assembly.

The cryo-EM/crystallography analysis is excellent, and the structural observations are very interesting. Considering that LSVs are only recently discovered, the work presents a vital groundwork that may prompt future functional studies to understand the roles of pH-induced particle dynamics, proteolysis of the C-termini, and RNA recruitment during virion assembly for LSV. However, there are some concerns:

Thank you very much for good comments and all the valuable suggestions to improve our paper.

1) The manuscript is exclusively structure-oriented, lacking any biochemical studies to correlate structural observations and substantiate the interpretations, which often look very speculative.

We have included UV spectroscopy for RNA encapsidation and mass spectroscopy for the mimicking autoproteolysis process of the C-terminus to enhance the study. We hope that our work may promote future functional studies of LSV and virion assembly.

2) The interpretation that 'trapezoid' capsomers are the 'domino-configured' building blocks, of the capsid assembly based on the observation of stringed features in cryo-EM images is highly speculative without further experimental evidence. It is quite possible that these features seen in the EM images have nothing to do with capsid assembly and could be dead-end products. Perhaps authors can include this figure in the supplementary material and suggest that these may be building blocks in the discussion.

Thank you for your comment on this issue. The trapezoid capsomeres with domino-scaffold configuration often appeared near incomplete or complete capsids in most of

our cryo-EM images (Supplementary Fig. 19b). These observations suggested that domino-scaffold configuration might be formed during the intermediate state of self-assembly of LSV capsids or during capsid disassembly, which leads to misassembled viral particles or dead-end pathway. However, the Reviewer is right that the trapezoid capsomeres might be the domino-configured building blocks or just off-path (or dead-end) products, which come from incorrect assembly or off-pathway. Our trapezoid capsomeres was newly observed, which might provide one possibility of assemble path or intermediate state. However, it requires more characterization in depth.

We follow Reviewer's suggestion to include this figure (original Fig. 6) in the supplementary figure (now Supplementary Fig. 19) and suggest that these may be building blocks and off-path products in the Discussion session. We also revised the session with sentences at the end: "On the other hand, some aberrant assemblies were reported to be more easily formed^{48,49}. It is thus also possible that trapezoid capsomeres might be off-path or dead-end products during assembly. The new observed trapezoid capsomeres require more characterization in depth in the future". (p. 14, para. 3, lines 10–12) The supplementary Fig. 19 is also revised with off-pathway products.

Reference:

- 48 Chevalier, C., Lepault, J., Da Costa, B. & Delmas, B. The last C-terminal residue of VP3, glutamic acid 257, controls capsid assembly of infectious bursal disease virus. *J. Virol.* 78, 3296–3303 (2004).
- 49 Medrano, M. et al. Imaging and quantitation of a succession of transient intermediates reveal the reversible self-assembly pathway of a simple icosahedral virus capsid. *J. Am. Chem. Soc.* 138, 15385–15396 (2016).

Supplementary Fig. 19b The domino-scaffold configuration appeared often near capsids in our cryo-EM images.

3) Similarly, the role of NTD in RNA packaging by observing only two bases in the structure of N-terminal deleted VLPs with T=3 symmetry is also highly speculative and unnecessary. Where did the RNA come from in the VLPs? Have the authors checked spectroscopically that these VLPs and not others contain RNA? Even if present, it has to be non-specific interaction and may not have anything to do with capsid assembly or packaging.

Thank you very much for pointing out this issue. The RNA might come from the expression host *E. coli* during expression and purification of capsid proteins. According to Reviewer's suggestion, we have performed the UV spectroscopy with absorbance at wavelengths 260 and 280 nm for the examination of the RNA content in LSV2 and delta-N48 LSV1 VLPs. The results show that the genomic RNA content from expression host *E. coli* for LSV2 and delta-N48 LSV1 VLPs could non-specifically incorporate with positively charged residues of LSV CPs.

We added the method of UV spectroscopy in the session of Methods and Materials, and presented the result in Supplementary Table 8. We also added sentences on Result: "The existence of nucleic acids inside LSV2 and delta-N48 LSV1 VLPs were further analyzed by measuring absorbance at 260 nm (A_{260}) and 280 nm (A_{280}), respectively. Utilizing these absorbance characteristics, a sum of the components can be calculated from the primary absorbance of the nucleic acid at A_{260} and the CPs at A_{280} after subtracting the background light scattering. The purified LSV2 and delta-N48 LSV1 VLPs showed the A_{260}/A_{280} ratio in the range of 1.68–1.88, respectively, indicating that both VLPs contain the nucleic acids (Supplementary Table 8)." (p. 8, para. 3, lines 1–6)

We also added sentences on Discussion and conclusions: "The A_{260}/A_{280} ratios of the LSV2 and delta-N48 LSV1 VLPs agree with the expected UV A_{260}/A_{280} ratio of RNA encapsidation by assembled VLPs *in vitro* of FHV⁴⁵. Several VLPs has been shown to predominantly package host RNA and its own encoding RNA^{46,47}. Our result indicates that the major genomic RNA content from the expression host *E. coli* and possible minor viral RNA2 of LSV2 and delta-N48 LSV1 CPs could incorporate with positively charged residues on some RNA-binding regions of LSV CPs (Supplementary Table 8). The additional asymmetric densities on the inner surface of all cryo-EM structures of LSV VLPs, might imply the locations, particularly at I5 axes, of the flexible RNA-binding regions and undefined RNA molecules (Fig. 3a and Supplementary Figs. 9a–b)." (p. 13, para. 2, lines 2–10).

From our data, this binding of RNA seems to be non-specific interaction. Thus, we remove all the statements of capsid assembly and packing related to RNA binding at this position, such as "We thus surmise that the disordered positively charged NTD coordinating with the nucleic acid around I3 (Q3) and I5 axes might play a crucial role in initializing and stabilizing the inter-capsomere contacts during capsid assembly"

Reference:

- 45 Bajaj, S. & Banerjee, M. In vitro assembly of polymorphic virus-like particles from the capsid protein of a nodavirus. *Virology* 496, 106–115 (2016).

- 46 Routh, A., Domitrovic, T. & Johnson, J. Host RNAs, including transposons, are encapsidated by a eukaryotic single-stranded RNA virus. *Proc. Natl. Acad. Sci.* **109**, 1907–1912 (2012).
- 47 Tetter, S. et al. Evolution of a virus-like architecture and packaging mechanism in a repurposed bacterial protein. *Science* **372**, 1220–1224 (2021).

4) *The crystal structure (at ~8 Å resolution) and the SAXS data do not add much to the manuscript. The changes in the LSV cryo-EM structures at different pHs are already quite convincing.*

Thank you for the comment. While cryo-EM structures of LSV at different pHs (6.5, 7.5 and 8.5) were measured, the crystal structure and the SAXS data not only support the changes of LSV structures but also provide the possible smallest diameter (crystal structure) and size transformation of LSV capsids with pH varied directly in column (SAXS). For that, we hope to keep both data in the manuscript but shorten the detailed descriptions.

“First, the $T=4$ LSV2 VLP crystals at resolution ~ 8 Å were grown under varied conditions with pH 6.5–7.0 (Fig. 5e and Supplementary Table 6). The $T=4$ particle size of diameter 450 Å based on the crystal packing is slightly smaller than the observed size range of the cryo-EM $T=4$ LSV2 VLP, suggesting the potential smallest shaped holes at the I2 (diameter 10 Å) and I5 (diameter 3.5 Å) axes (Figs. 5a–b). This smallest diameter of the unexpanded LSV2 particle is larger than that of tetraviruses (Supplementary Fig. 17). Notwithstanding this difference of $T=4$ particle sizes, the crystal and cryo-EM structures of $T=4$ LSV2 VLP share the similar CP conformation.” (p. 10, para. 2, lines 2–8)

“Second, for SAXS, the pH environments of LSV2 VLP were directly transformed from pH 7.5 to pH 6.5 and 8.5 in column, respectively; the full sizes of $T=4$ and $T=3$ LSV2 VLPs at pH 6.5 are estimated as ~ 482 and 438 Å, respectively, whereas those of $T=4$ and $T=3$ LSV2 VLPs at pH 8.5 increased to 494 and 449 Å, respectively (Fig. 5f and Supplementary Table 7).” (p. 10, para. 3, lines 1–4)

5) The statement in one of the sentences in the introduction, “presumably belonging to betatetraviruses of the Tetraviridae family with characteristic $T=4$ quasi-equivalence nonenveloped capsids,” is not well supported and perhaps misleading. It is yet unclear whether native infectious LSV viruses have $T=3$ capsid symmetry or $T=4$ symmetry. Either the authors should include a supporting citation or remove this part of the sentence.

Thank you for raising this issue. It is indeed unclear yet whether native infectious LSV viruses exhibit $T=3$ or $T=4$ capsid symmetry. A recent study classified LSV into a new family *Sinhaliviridae* within order *Nodamuvirales*. We removed the misleading sentence.

In the introduction, we have revised the sentences: “Honeybee-infecting LSV is currently classified into a new family *Sinhaliviridae* within order *Nodamuvirales*^{14,15}; however, its capsid gene and monopartite genome structure presumably are closer to betatetraviruses of the family *Alphatetraviridae* and Providence virus (PrV) of the family *Carmotetraviridae* with a characteristic $T=4$ quasi-equivalence nonenveloped

capsids packaging the single-stranded positive-sense RNA genome^{10,16,17}.” (p. 3 para. 1, lines 1–5)

In Discussion, we also revised the sentences: “Nevertheless, the characteristics of tetraviruses not only have a very restricted host range that is limited to lepidopteran insects but also reflect nonenveloped icosahedral $T=4$ virions with a diameter of ~42 nm containing either monopartite or bipartite positive-sense ssRNA genomes, similar to LSVs in our study. As above mentioned, we anticipate that LSVs, currently classified into family *Sinhaliviridae*, belong to the new genus of tetraviruses with $T=4$ virions between the families *Alphatetraviridae*, *Carmotetraviridae* and *Nodaviridae* that generally infect the larvae of insect species^{17,23,24,27,28,33,34,39}.” (p. 11 para. 3 lines 9–14)

Reference:

- 10 Daughenbaugh, K. et al. Honey bee infecting lake sinai viruses. *Viruses*. 7, 3285–3309 (2015).
- 14 Runckel, C. et al. Temporal analysis of the honey bee microbiome reveals four novel viruses and seasonal prevalence of known viruses, nosema, and crithidia. *PLoS One* 6, e20656 (2011).
- 15 Walker, P. et al. Changes to virus taxonomy and the statutes ratified by the international committee on taxonomy of viruses. *Arch. Virol.* 165, 2737–2748 (2020).
- 16 Hanzlik, T., Gordon, K., Maramorosch, K., Murphy, F. & Shatkin, A. The Tetraviridae. *Adv. Virus Res.* 48, 101–168 (1997).
- 17 Koonin, E., Dolja, V. & Krupovic, M. Origins and evolution of viruses of eukaryotes: The ultimate modularity. *Virology* 479, 2–25 (2015).
- 23 Helgstrand, C., Munshi, S., Johnson, J. & Liljas, L. The refined structure of Nudaurelia capensis omega Virus reveals control elements for a $T=4$ capsid maturation. *Virology* 318, 192–203 (2004).
- 24 Tang, L. et al. The structure of Pariacoto virus reveals a dodecahedral cage of duplex RNA. *Nat. Struct. Biol.* 8, 77–83 (2001).
- 27 Cheng, R. et al. Functional implications of quasi-equivalence in a $T=3$ icosahedral animal virus established by cryoelectron microscopy and x-ray crystallography. *Structure* 2, 271–282 (1994).
- 28 Fisher, A. & Johnson, J. Ordered duplex RNA controls capsid architecture in an icosahedral animal virus. *Nature* 361, 176–179 (1993).
- 33 Domitrovic, T., Matsui, T. & Johnson, J. Dissecting quasi-equivalence in nonenveloped viruses: membrane disruption is promoted by lytic peptides released from subunit pentamers, not hexamers. *J. Virol.* 86, 9976–9982 (2012).
- 34 Speir, J. A. et al. Evolution in action: N and C termini of subunits in related $T=4$ viruses exchange roles as molecular switches. *Structure* 18, 700–709 (2010).
- 39 Schneemann, A. & Marshall, D. Specific encapsidation of nodavirus RNAs is mediated through the C terminus of capsid precursor protein alpha. *J. Virol.* 72, 8738–8746 (1998).

6) In the sentence “The models of $T=4$ and $T=3$ LSV2 VLPs were built and refined with the homo-trimer A/B/C or D/D/D in one asymmetric unit, which form oligomerizes into a homo-tetrameric and a homo-trimeric capsomeres”, the part “which form oligomerizes into homo-tetrameric and homo-trimeric capsomeres” is speculative and unnecessary in

this sentence. The authors later suggest that the capsid subunits form homo-trimeric or homotetrameric capsomere. Still, it is unnecessary and somewhat confusing to include it as a part of the above sentence.

Thank you very much for pointing this out. We have deleted the part “which form oligomerizes into homo-tetrameric and homo-trimeric capsomeres” as well as other descriptions related to capsid subunits forming homo-trimeric or homo-tetrameric capsomeres in the manuscript to avoid the speculative and unnecessary statements. (p.4, para. 2, lines 2)

7) The description of the structural features in several places can be succinct and less confusing.

Thank you. We have extensively and carefully amended and revised the manuscript to be succinct and less confusing.

8) Overall, the manuscript can be substantially improved by 1) restricting the text to the description of structural features and avoiding unnecessary speculations, 2) revising the subtitles (some of which are redundant) to reflect more appropriately the contents that follow, and 3) removing repetitions of the same idea that occur in several places.

Thank you for the comments for the improvement of our manuscript. We have restricted the text to the description of structural features and removed or shortened the unnecessary speculations. We also revised the subtitles and removed the repetitions throughout the manuscript. Now the paper read much succinct and less confusing. Thank you very much.

9) “Long spikes (length 3.6 nm) protrude from the center of each trimeric capsomeres at icosahedral 3-fold (I3) or Q3 axes” Should it be 3-fold (I3) and Q3 axes? In the T=4 structure, the spikes are located at both icosahedral and quasi 3-fold axes, whereas in the T=3 structures, they are located at the quasi 3-fold axes.

We appreciate the Reviewer for pointing this out. Yes, it should be icosahedral 3-fold (I3) and Q3 axes. We have corrected the sentence as “Long spikes (length 3.6 nm) protrude from the center of each homo-trimeric capsomeres at icosahedral 3-fold (I3) and quasi 3-fold (Q3) axes in one T=4 capsid, and at Q3 axes in one T=3 capsid.” (p. 4, para. 2, lines 6–8)

11) Several typos noted

Thank you for pointing it out. We have carefully read and corrected all the typos throughout the manuscript.

Reviewer #2

In their manuscript, Chen et al. present several viral like particles (VLP) structures formed by the capsid protein (CP) of Honeybee Lake Sinai Virus (LSV1 and LSV2). These viruses have a single-stranded positive sense RNA genome and cause colony collapse disorder in apiaries and have a T=4 icosahedral capsid.

The authors expressed the CP proteins of LSV1 (residues 49-565) and LSV2 (residues 1-520 i.e. the full length protein) in E. Coli. Upon dialysis against VLP-forming buffers (at pH 6.5, 7.5 and 8.5), purified CPs self assemble into VLPs with T=3 or T=4 icosahedral symmetry.

The authors determine the structure of T=4 and T=3 LSV2 VLPs at pH 7.5, 8.5 and 6.5 and T=4 and T=3 delta-N48 LSV1 VLPs at pH 6.5, using cryo-electron microscopy (cryo-EM) combined with X-ray crystallography.

They present a detailed description of LSP2 assemblies and compare them with T=4 Nudaurelia capensis ω virus, T=3 pariacoto virus and T=1 infectious bursal disease virus assemblies, which all also have a central beta-barrel domain. They identify an anchoring loop, involved in both the stability of T3 and T4 VLP, while N ω V, PrV and PaV ensure the stability of their capsid with a calcium ion located at the same place as LSV loop.

Previous studies have demonstrated that PrV CP and N ω V CP undergo autoproteolytic cleavage. The authors provide evidence that it is not the case for LSV. However, a double mutation (M83E/D461F) introduced into the LSV2 CP, restoring the autoproteolysis sites of PrV and N ω V, seems to trigger autocleavage of LSV2 CP.

In the refined reconstruction of LSV1 VLP, they observe a two-based single-strand RNA indicative of a genome binding site.

Finally they also observe a compaction of the capsid at low pH.

While I acknowledge the mass of data provided, I am not sure this detailed work will be of interest to researchers beyond the concerned community. As examples, structures of Nudaurelia capensis omega Virus were published in J. Virol in 2002, in Virology in 2004, in Protein Science in 2005, etc.

In any case, I have some comments that should be taken into account.

Thank you very much for the good comments and suggestions. We have revised and re-organized the main text to improve the presentation of the manuscript.

Major points:

1) In their analysis of the profile of the size exclusion chromatography presented in figure 2C (p7, line 6), the authors mention that the deletion of the anchor segment of 13 residues induces a substantial decrease of the amount of LSV2 VLPs. However they did not comment the broad peak of proteins eluting between ~15 and ~22 ml which exhibits several shoulders. It is unclear why they attribute one of these shoulders more than the other to

monomeric CP. An SDS PAGE gel of the sample before size exclusion would be useful (to see the purity of the proteins) as well as an analysis of the different fractions by SDS PAGE and negative staining.

Thank you for pointing this out. We have repeatedly and cautiously purified LSV2 anchor loop deletion mutant by SEC for further analyses with SDS-PAGE and negative-staining EM as suggested by Reviewer (Fig. 2c and Supplementary Fig. 13). First, we observed by SEC that the major proportion of the LSV2 anchor loop deletion mutant failed to assemble VLPs and remained intermediates that are smaller than VLPs. The SDS-PAGE analysis showed that the shoulder and the wider profile of the peak 2 were caused by the several un-pure proteins contaminated with the anchor loop deletion mutant, whereas the peak 1 with the sharp profile contained only the anchor loop deletion mutant. Second, the analysis of the negative-staining EM images indicated that the pooled fractions of peak 1 had the propensity of self-interaction, since protein aggregates, as well as a few irregular particles. Thus, unlike full-length LSV2 CP, which assemble into capsids, the pooled fractions of peak 2 assemble into smaller aggregations, confirming that this newly identified anchor loop is functional for viral capsid assembly.

We have added the analyses of SDS PAGE and negative-staining EM in the main text and Supplementary Fig. 13.

“Intriguingly, a deletion mutant without the anchor segment of 13 residues (356–378) substantially decreases the yield of intact LSV2 VLP (Fig. 2c). With size-exclusion chromatography (SEC), we observed that the major proportion of the LSV2 anchor-loop-deletion mutant generally failed to assemble VLPs and remained intermediates that were smaller than the complete VLPs. The fractions of peak 1 and peak 2 were further pooled and isolated with SEC and analyzed with SDS-PAGE and negative-staining EM (Supplementary Fig. 13). As a result, the peak 1, containing a mixture of larger CP aggregations, irregular particles and only few VLPs, had the propensity for self-interaction without assembly order, whereas the peak 2, with smaller variable intermediates of CP, lost the assembly ability. Compared to full-length LSV2 CPs tending to assemble into complete capsids, the behavior of the deletion mutant confirmed that this newly identified anchor loop is functional for viral capsid assembly.” (p. 6 para. 5, lines 1–10)

“The mutant with anchor loop deletion of LSV2 CP greatly reduced the VLP formation of the first peak fraction in SEC through analyses of SDS-PAGE and negative-staining EM (Fig. 2c and Supplementary Fig. 13); intermediates were predominantly observed in the second peak fraction owing to deficient hydrogen bonds from anchor loops. The anchor loop is thus necessary to perform the formation of the homo-trimeric capsomeres in an earlier stage of LSV capsid assembly.” (p. 11 para. 2 Lines 4–8)

2) Similarly, the gel presented in figure 3C only suggests that there is a cleavage. The precise site could be analyzed by mass spectrometry. Incidentally, one can notice that the CP proteins although the majority in the gel, are not so pure which reinforces the need of a more in-depth analysis in my previous remark.

Thank you for the raising this issue. In the original Fig. 3c, we tried to show the differences of characteristic autoproteolysis process on the major bands between the full-length and M83E/D461F LSV2 CPs. The original SDS PAGE gel looked impure with some contaminations. These two CPs were only purified from the IMAC columns and were treated with SUMO protease. Therefore, these two specimens included our target CPs, SUMO tag and some minor *E. coli* contaminated proteins. We have purified the CPs of full-length and M83E/D461F LSV2 with the Superdex 200 Increase 10/300 GL and Superose 6 10/300 GL columns to remove SUMO tag and those minor contaminations. The new SDS-PAGE gel is now shown in Fig. 4c. We are sorry for our previous SDS-PAGE presentation to cause any misunderstanding. Thank you for pointing it out and letting us to improve our presentation.

Furthermore, we have performed the MALDI-TOF mass spectrometry experiment, and results showed that M83E/D461F in a neutral pH buffer exhibited the autoproteolytic capability and the cleavage site was at the scissile peptide bond Y459–N460. We have included the new data, a new supplementary figure (Supplementary Fig. 16), result and discussion of the mimicking autoproteolysis process in the manuscript in details.

In the result: “The mutant M83E/D461F migrated on SDS-PAGE with a band corresponding to a molecular weight less than that of full-length CP by ~7 kDa (Fig. 4c). Furthermore, we confirmed that each band on SDS-PAGE corresponding to the full-length LSV2 CP and the M83E/D461F by in-gel trypsinolysis followed by MALDI-TOF MS-based proteomics (Supplementary Fig. 16). Analyses of the multiple peptides from these two bands result in 79 and 75% sequence coverages for 520 amino acids of LSV2 CP and the M83E/D461F, respectively. As a result, first, two peptides (residues 501–511 and 512–520) consonant with the last 20 residues of the C-terminus was detected only from the full-length LSV2 CP but not from M83E/D461F. Second, an additional peak at 1247.564 m/z was observed only in M83E/D461F and conformed the specific peptide mass of residues 449–459). The MALDI-TOF MS analyses suggest that the M83E/D461F possesses autoproteolysis at the cleavage site Y459–N460 on the scissile bond.” (p. 9 para. 2 Lines 3–12)

In the discussion: “Mass spectrum analysis further showed that the cleavage site is between Y459 and N460 (Supplementary Fig. 16), presumably releasing a C-terminus peptide. This autoproteolytic activity might be achieved when the catalytic M83E attacks the scissile peptide bond (Y459–N460) that was moved structurally closer to the M83E residue while D461 was mutated to Phe. Further investigations are needed to elucidate the mechanism of autoproteolysis cleavage with the associated catalytic residues and the scissile bond. All above differences in dynamic particle size with pH variation and C-termini between the various strains of LSV and tetraviruses might imply that the C-termini of LSV CPs exhibit other functional properties different from the conventional multifunction of the γ peptide. Further investigation is needed to determine the factors that controlling the efficiency of the quaternary rearrangements that accompany maturation or viral genome releasing in LSV.” (p. 13 para. 1 Lines 2–11)

3) *The authors suggest that the trapezoid-shape capsomeres that they observe in figure 6A constitute the basis of VLP architecture. They may be right but these objects may also*

correspond to off-path (and even dead-end path) assemblies. Particularly, out of the infectious context, it is known that some aberrant assemblies are more easily formed (e.g. Chevalier et al. *J Virol.* 2004. 78: 3296-3303. doi: 10.1128/jvi.78.7.3296-3303.2004).

Thank you for your comment on this issue. The trapezoid capsomeres were observed in several cryo-EM images. The Reviewer is right that the trapezoid capsomeres might be the domino-configured building blocks or just off-path (or dead-end) products. It is indeed that some aberrant assemblies are more easily formed as Chevalier et al. observed VLPs, tubes, irregular particles and disk-like structures. Our trapezoid capsomeres was newly observed, which might provide one possibility of assemble path or intermediate state. However, it requires more characterization in depth. We now follow the Reviewer 1's suggestion and your comment to include this figure (original Fig. 6) in the supplementary materials (Supplementary Fig. 19) and suggest that these may be building blocks or off-path product in the Discussion session. We also revised the session with sentences at the end: "On the other hand, some aberrant assemblies were reported to be more easily formed^{48,49}. It is thus also possible that trapezoid capsomeres might be off-path or dead-end products during assembly. The new observed trapezoid capsomeres require more characterization in depth in the future". The supplementary Fig. 19 is also revised with off-pathway products. (p. 14, para. 3, lines 10–12)

Reference:

- 48** Chevalier, C., Lepault, J., Da Costa, B. & Delmas, B. The last C-terminal residue of VP3, glutamic acid 257, controls capsid assembly of infectious bursal disease virus. *J. Virol.* 78, 3296–3303 (2004).
- 49** Medrano, M. et al. Imaging and Quantitation of a succession of transient intermediates reveal the reversible self-assembly pathway of a simple icosahedral virus capsid. *J. Am. Chem. Soc.* 138, 15385–15396 (2016).

Supplementary Fig. 19b The domino-scaffold configuration appeared often near capsids in our cryo-EM images.

Minor points:

1) The authors should correct the presentation of the taxonomy in the introduction: the *Tetraviridae* family has been dissolved in 2011 and replaced by three families (*Alphatetraviridae*, *Carmotetraviridae* and *Permutotetraviridae*). The genera *Omegatetravirus*, and *Betatetravirus* belong to the *Alphatetraviridae* family.

Thank you for the good suggestion and information. We have corrected the presentation of the taxonomy with classification and distinction of our LSV and tetraviruses in the introduction and main text accordingly. LSV is currently classified into a new family *Sinhaliviridae* within order *Nodamuvirales*. We also changed the term *Tetraviridae* to tetraviruses and revised the descriptions.

“Honeybee-infecting LSV is currently classified into a new family *Sinhaliviridae* within order *Nodamuvirales*^{14,15}; however, its capsid gene and monopartite genome structure presumably are closer to betatetraviruses of the family *Alphatetraviridae* and Providence virus (PrV) of the family *Carmotetraviridae* with a characteristic $T=4$ quasi-equivalence nonenveloped capsids packaging the single-stranded positive-sense RNA genome^{10,16,17}.” (p. 3 para. 1, lines 1–5)

“LSV1 and LSV2 CPs with molecular masses ~63 and 57 kDa, respectively, share low sequence similarities with other known viruses of families *Dicistroviridae*, *Flaviridae* and *Nodaviridae* as well as tetraviruses.” (p. 3, para. 2, lines 4–6)

“Nevertheless, the characteristics of tetraviruses not only have a very restricted host range that is limited to lepidopteran insects but also reflect nonenveloped icosahedral $T=4$ virions with a diameter of ~42 nm containing either monopartite or bipartite positive-sense ssRNA genomes, similar to LSVs in our study. As above mentioned, we anticipate that LSVs, currently classified into family *Sinhaliviridae*, belong to the new genus of tetraviruses with $T=4$ virions between the families *Alphatetraviridae*, *Carmotetraviridae* and *Nodaviridae* that generally infect the larvae of insect species^{17,23,24,27,28,33,34,39}.” (p. 11 para. 3 lines 9–14)

Reference:

- 10 Daughenbaugh, K. et al. Honey bee infecting lake sinai viruses. *Viruses*. 7, 3285–3309 (2015).
- 14 Runckel, C. et al. Temporal analysis of the honey bee microbiome reveals four novel viruses and seasonal prevalence of known viruses, nosema, and crithidia. *PLoS One* 6, e20656 (2011).
- 15 Walker, P. et al. Changes to virus taxonomy and the statutes ratified by the international committee on taxonomy of viruses. *Arch. Virol.* 165, 2737–2748 (2020).
- 16 Hanzlik, T., Gordon, K., Maramorosch, K., Murphy, F. & Shatkin, A. The Tetraviridae. *Adv. Virus Res.* 48, 101–168 (1997).
- 17 Koonin, E., Dolja, V. & Krupovic, M. Origins and evolution of viruses of eukaryotes: The ultimate modularity. *Virology* 479, 2–25 (2015).
- 23 Helgstrand, C., Munshi, S., Johnson, J. & Liljas, L. The refined structure of Nudaurelia capensis omega Virus reveals control elements for a $T=4$ capsid maturation. *Virology* 318, 192–203 (2004).
- 24 Tang, L. et al. The structure of Pariacoto virus reveals a dodecahedral cage of duplex RNA. *Nat. Struct. Biol.* 8, 77–83 (2001).
- 27 Cheng, R. et al. Functional implications of quasi-equivalence in a $T=3$ icosahedral animal virus established by cryoelectron microscopy and x-ray crystallography. *Structure* 2, 271–282 (1994).
- 28 Fisher, A. & Johnson, J. Ordered duplex rna controls capsid architecture in an icosahedral animal virus. *Nature* 361, 176–179 (1993).
- 33 Domitrovic, T., Matsui, T. & Johnson, J. Dissecting quasi-equivalence in nonenveloped viruses: membrane disruption is promoted by lytic peptides released from subunit pentamers, not hexamers. *J. Virol.* 86, 9976–9982 (2012).
- 34 Speir, J. A. et al. Evolution in action: N and C termini of subunits in related $T=4$ viruses exchange roles as molecular switches. *Structure* 18, 700–709 (2010).
- 39 Schneemann, A. & Marshall, D. Specific encapsidation of nodavirus RNAs is mediated through the C terminus of capsid precursor protein alpha. *J. Virol.* 72, 8738–8746 (1998).

2) PrV should be identified as Providence Virus somewhere in the text.

Thank you for pointing this out. We have mentioned and defined PrV as Providence Virus at its first appearance in the text: “Providence virus (PrV)”. (p. 3, para. 1, line 3)

3) Several typos and misspelled word in the text which should be corrected.

Thank you for pointing it out. We have carefully read and corrected the typos and misspelled word throughout the manuscript.

Reviewer #1 (Remarks to the Author):

The authors have addressed my concerns/comments/suggestions adequately. Although I do not have any further comments, the manuscript needs more editorial revisions for clarity in language. For example, the subtitle "Structural differences of T=4 and T=3 LSV2 VLPs" needs to be changed to "Structural differences between T=4 and T=3 LSV2 VLPs"

Reviewer #2 (Remarks to the Author):

In their revised version of the manuscript which describes the atomic structures of virus-like particles (VLPs) of Lake Sinai virus (LSV) determined using cryo-EM and crystallography, the authors have taken my previous comments into account.

They have included new data (UV spectroscopy to investigate RNA encapsidation and mass spectroscopy to characterize the mimicking autoproteolysis process of LSV2 C-terminus).

They have also toned down some of their earlier interpretations.

Globally, the manuscript is much better and this detailed structural work will be of interest to researchers in the field who wish to develop functional studies.

I have two minor comments :

p13 The authors wrote: « Our result indicates that the major genomic RNA content from the expression host E. coli and possible minor viral RNA2 of LSV2 and delta-N48 LSV1 CPs could incorporate with positively charged residues on some RNA-binding regions of LSV CPs ». The meaning of this sentence is not very clear to me. As it is difficult to further discuss the origin and nature of this RNA in the absence of sequencing data, the authors could simply write that: « Our results suggest that capsids can incorporate E Coli RNA which probably interacts with positively charged residues on some RNA-binding regions of LSV CPs. »

Bottom of p6 The authors wrote: « the behavior of the deletion mutant confirmed that this newly identified anchor loop is functional for viral capsid assembly. » I would rather write « the behavior of the deletion mutant confirmed that this newly identified anchor loop is required for proper viral capsid assembly. »

POINT-BY-POINT responses to the comments of the Reviewers

Reviewer #1

The authors have addressed my concerns/comments/suggestions adequately. Although I do not have any further comments, the manuscript needs more editorial revisions for clarity in language. For example, the subtitle "Structural differences of T=4 and T=3 LSV2 VLPs" needs to be changed to "Structural differences between T=4 and T=3 LSV2 VLPs"

Thank very much for your review of our manuscript. The valuable and constructive comments and suggestions greatly improve our manuscript and studies. We have carefully revised the language and grammar throughout the manuscript with reading and editing by two colleagues.

Reviewer #2

In their revised version of the manuscript which describes the atomic structures of virus-like particles (VLPs) of Lake Sinai virus (LSV) determined using cryo-EM and crystallography, the authors have taken my previous comments into account. They have included new data (UV spectroscopy to investigate RNA encapsidation and mass spectroscopy to characterize the mimicking autoproteolysis process of LSV2 C-terminus). They have also toned down some of their earlier interpretations. Globally, the manuscript is much better and this detailed structural work will be of interest to researchers in the field who wish to develop functional studies.

We are grateful to the reviewer. The valuable and constructive comments and suggestions greatly improve our manuscript and studies.

I have two minor comments:

p13 The authors wrote: « Our result indicates that the major genomic RNA content from the expression host E. coli and possible minor viral RNA2 of LSV2 and delta-N48 LSV1 CPs could incorporate with positively charged residues on some RNA-binding regions of LSV CPs ». The meaning of this sentence is not very clear to me. As it is difficult to further discuss the origin and nature of this RNA in the absence of sequencing data, the authors could simply write that: « Our results suggest that capsids can incorporate E Coli RNA which probably interacts with positively charged residues on some RNA-binding regions of LSV CPs. »

Thank you. We have revised this sentence as: “Our results suggest that capsids can incorporate *E. coli* RNA which probably interacts with positively charged residues on some RNA-binding regions of LSV CPs.” (p. 13, para. 2, lines 5–6)

Bottom of p6 The authors wrote: « the behavior of the deletion mutant confirmed that this newly identified anchor loop is functional for viral capsid assembly. » I would rather write « the behavior of the deletion mutant confirmed that this newly identified anchor loop is required for proper viral capsid assembly.

We have edited the sentence as “the behavior of the deletion mutant confirmed that this newly identified anchor loop is required for proper viral capsid assembly.” (p. 6, para. 4, lines 9–10)